# OpenVLA:
# An Open-Source Vision-Language-Action Model

**Moo Jin Kim**[*,1]   **Karl Pertsch**[*,1,2]   **Siddharth Karamcheti**[*,1,3]

**Ted Xiao**[4]   **Ashwin Balakrishna**[3]   **Suraj Nair**[3]   **Rafael Rafailov**[1]   **Ethan Foster**[1]

**Pannag Sanketi**[4]   **Quan Vuong**[5,†]   **Thomas Kollar**[3]   **Benjamin Burchfiel**[3]   **Russ Tedrake**[3,6]   **Dorsa Sadigh**[1]

**Sergey Levine**[2]   **Percy Liang**[1]   **Chelsea Finn**[1]

https://openvla.github.io

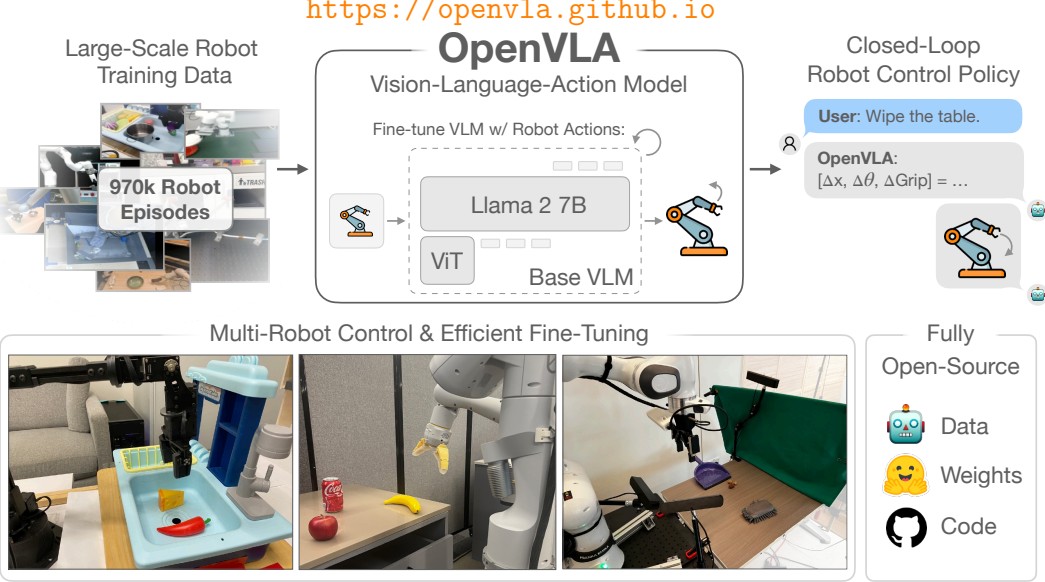

Figure 1: We present OpenVLA, a 7B-parameter open-source vision-language-action model (VLA), trained on 970k robot episodes from the Open X-Embodiment dataset [1]. OpenVLA sets a new state of the art for generalist robot manipulation policies. It supports controlling multiple robots out of the box and can be quickly adapted to new robot domains via parameter-efficient fine-tuning. The OpenVLA checkpoints and PyTorch training pipeline are fully open-source and models can be downloaded and fine-tuned from HuggingFace.

**Abstract:** Large policies pretrained on a combination of Internet-scale vision-language data and diverse robot demonstrations have the potential to change how we teach robots new skills: rather than training new behaviors from scratch, we can fine-tune such vision-language-action (VLA) models to obtain robust, generalizable policies for visuomotor control. Yet, widespread adoption of VLAs for robotics has been challenging as 1) existing VLAs are largely closed and inaccessible to the public, and 2) prior work fails to explore methods for efficiently fine-tuning VLAs for new tasks, a key component for adoption. Addressing these challenges, we introduce OpenVLA, a 7B-parameter open-source VLA trained on a diverse collection of 970k real-world robot demonstrations. OpenVLA builds on a Llama 2 language model combined with a visual encoder that fuses pretrained features from DINOv2 and SigLIP. As a product of the added data diversity and new model components, OpenVLA demonstrates strong results for generalist manipulation, outperforming closed models such as RT-2-X (55B) by 16.5% in absolute task success rate across 29 tasks and multiple robot embodiments, with 7x fewer parameters. We further show that we can effectively fine-tune OpenVLA for new settings,

---

*: denotes equal contribution

Correspondence to: moojink@stanford.edu, pertsch@berkeley.edu, skaramcheti@stanford.edu

[1]Stanford University, [2]UC Berkeley, [3]Toyota Research Institute, [4]Google Deepmind, [5]Physical Intelligence, [6]MIT, [†]Work done in part while at Google Deepmind

8th Conference on Robot Learning (CoRL 2024), Munich, Germany.

with especially strong generalization results in multi-task environments involving multiple objects and strong language grounding abilities, where we outperform expressive from-scratch imitation learning methods such as Diffusion Policy by 20.4%. We also explore compute efficiency; as a separate contribution, we show that OpenVLA can be fine-tuned on consumer GPUs via modern low-rank adaptation methods and served efficiently via quantization without a hit to downstream success rate. Finally, we release model checkpoints, fine-tuning notebooks, and our PyTorch codebase with built-in support for training VLAs at scale on Open X-Embodiment datasets.

**Keywords:** Vision-Language-Action Models, Generalist Policies, Large-scale Robot Learning

# 1 Introduction

A key weakness of learned policies for robotic manipulation is their inability to generalize beyond their training data: while existing policies trained for individual skills or language instructions have the capacity to extrapolate behaviors to new initial conditions such as object positions or lighting [2, 3], they lack robustness to scene distractors or novel objects [4, 5] and struggle to execute unseen task instructions [6, 7]. Yet beyond robotics, existing foundation models for vision and language such as CLIP [8], SigLIP [9], and Llama 2 [10] are capable of these types of generalization and more, stemming from the priors captured by their Internet-scale pretraining datasets. While reproducing this scale of pretraining for robotics is still an open challenge — even the largest robot manipulation datasets [1, 11] only have 100K to 1M examples – this imbalance suggests an opportunity: using existing foundation models for vision and language *as a core building block* for training robotic policies that can generalize to objects, scenes, and tasks beyond their training data.

Towards this goal, existing work has explored integrating pretrained language and vision-language models for robotic representation learning [12–14] and as a component in modular systems for task planning and execution [15, 16]. More recently, they have been used for directly learning vision-language-action models [VLAs; 1, 7, 17, 18] for control. VLAs provide a direct instantiation of using pretrained vision-and-language foundation models for robotics, directly fine-tuning visually-conditioned language models (VLMs) such as PaLI [19, 20] to generate robot control actions. By building off of strong foundation models trained on Internet-scale data, VLAs such as RT-2 [7] demonstrate impressive robustness results, as well as an ability to generalize to novel objects and tasks, setting a new standard for generalist robot policies. Yet, there are two key reasons preventing the widespread use of existing VLAs: 1) current models [1, 7, 17, 18] are *closed*, with limited visibility into model architecture, training procedures, and data mixture, and 2) existing works do not provide best practices for *deploying and adapting VLAs* to new robots, environments, and tasks — especially on commodity hardware (e.g., consumer-grade GPUs). To develop a rich foundation for future research, robotics needs open-source, generalist VLAs that support effective fine-tuning and adaptation, akin to the existing ecosystem around open-source language models [21–24].

To this end, we introduce OpenVLA, a 7B-parameter open-source VLA that establishes a new state of the art for generalist robot manipulation policies. OpenVLA consists of a pretrained visually-conditioned language model backbone that captures visual features at multiple granularities, fine-tuned on a large, diverse dataset of 970k robot manipulation trajectories from the Open-X Embodiment [1] dataset — a dataset that spans a wide range of robot embodiments, tasks, and scenes. As a product of increased data diversity and new model components, OpenVLA outperforms the 55B-parameter RT-2-X model [1, 7], the prior state-of-the-art VLA, by 16.5% absolute success rate across 29 evaluation tasks on the WidowX and Google Robot embodiments. We additionally investigate *efficient fine-tuning strategies for VLAs*, a new contribution not explored in prior work, across 7 diverse manipulation tasks spanning behaviors from object pick-and-place to cleaning a table. We find that fine-tuned OpenVLA policies clearly outperform fine-tuned pretrained policies such as Octo [5]. Compared to from-scratch imitation learning with diffusion policies [3], fine-tuned OpenVLA shows substantial improvement on tasks involving grounding language to behavior in multi-task settings with multiple objects. Following these results, we are the first to demonstrate the effectiveness

of compute efficient fine-tuning methods leveraging low-rank adaptation [LoRA; 25] and model quantization [26] to facilitate adapting OpenVLA models on consumer-grade GPUs instead of large server nodes without compromising performance. As a final contribution, we open-source all models, deployment and fine-tuning notebooks, and the OpenVLA codebase for training VLAs at scale, with the hope that these resources enable future work exploring and adapting VLAs for robotics.

## 2 Related Work

**Visually-Conditioned Language Models** Visually-conditioned language models (VLMs), which are trained on Internet-scale data to generate natural language from input image(s) and language prompts, have been adopted for myriad applications from visual question answering [27–30] to object localization [31, 32]. VLMs are typically models that bridge features from pretrained vision encoders [8, 9, 33] with pretrained language models [10, 23, 34–36]. While early work explored various architectures for cross-attending between vision and language features [37–41], new open-source VLMs [20, 42–44] have converged on a simpler "patch-as-token" approach, in which patch features from pretrained visual transformers are treated as tokens, and are then projected into the input space of a language model. These models build the backbone of our OpenVLA policy.

**Generalist Robot Policies** A recent trend in robotics works towards training multi-task "generalist" robot policies [2, 6, 45–49] on large diverse robot datasets [1, 2, 6, 11, 45, 49–56], spanning many different robot embodiments [1, 5, 53, 57–66]. Notably, Octo [5] trains a generalist policy that can control multiple robots out-of-the-box and allows for flexible fine-tuning to new robot setups. A key difference between these approaches and OpenVLA is the model architecture. Prior works like Octo typically compose pretrained components such as language embeddings or visual encoders with additional model components initialized from scratch [2, 5, 6], learning to "stitch" them together during the course of policy training. OpenVLA adopts a more end-to-end approach, directly fine-tuning visually-conditioned language models to generate robot actions by treating them as tokens in the language model vocabulary. Our experimental evaluation shows that this simple yet scalable pipeline substantially boosts performance and generalization ability over prior generalist policies.

**Vision-Language-Action Models** A number of works have explored the use of VLMs for robotics, e.g., for visual state representations [12, 13], object detection [67], high-level planning [16], and for providing a feedback signal [68–71]. Others integrate VLMs directly into end-to-end visuomotor manipulation policies [14, 15], but incorporate significant structure into the policy architecture or require calibrated cameras. Multiple works directly fine-tune large pretrained VLMs for predicting robot actions [1, 7, 17, 18, 72–74]. Such models are often referred to as vision-language-action models (VLAs), since they fuse robot control actions directly into VLM backbones. Existing works on VLAs either train and evaluate in single robot or simulated setups [72–75] and thus lack generality, or are closed and do not support efficient fine-tuning to new robot setups [1, 7, 17, 18]. Most closely related, RT-2-X [1] trains a 55B-parameter VLA policy on the Open X-Embodiment dataset [1] and demonstrates state-of-the-art generalist manipulation policy performance. However, our work differs from RT-2-X in multiple important aspects: (1) by combining a strong open VLM backbone with a richer robot pretraining dataset, OpenVLA matches or outperforms RT-2-X in our experiments while being 7x smaller; (2) we thoroughly investigate fine-tuning of OpenVLA models to new target setups, while RT-2-X does not investigate the fine-tuning setting; (3) we are the first to demonstrate the effectiveness of modern parameter-efficient fine-tuning and quantization approaches for VLAs; and (4) OpenVLA is the first generalist VLA that is open-source and thus supports future research on VLA training, data mixtures, objectives, and inference.

## 3 The OpenVLA Model

We introduce the OpenVLA model, a 7B-parameter vision-language-action model (VLA) trained on 970k robot demonstrations from the Open X-Embodiment dataset [1]. In this section, we first provide a brief overview of modern VLMs, which form the backbone of OpenVLA (Section 3.1), describe our training recipe (Section 3.2) and dataset (Section 3.3), and discuss key design decisions (Section 3.4). Details of the infrastructure used for training and inference are in Appendix C, and a description of our codebase is in Appendix D.

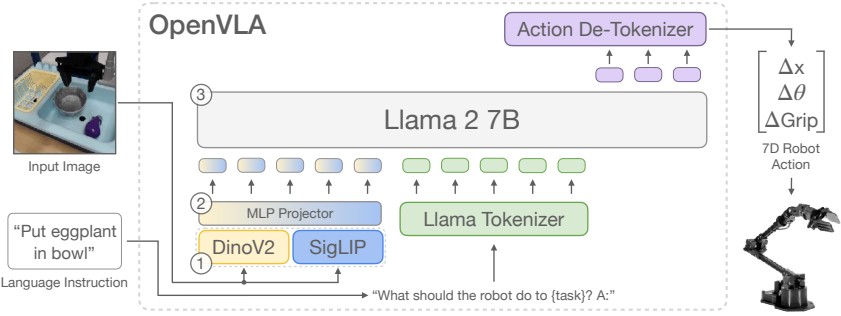

Figure 2: **OpenVLA model architecture.** Given an image observation and a language instruction, the model predicts 7-dimensional robot control actions. The architecture consists of three key components: (1) a **vision encoder** that concatenates Dino V2 [33] and SigLIP [76] features, (2) a **projector** that maps visual features to the language embedding space, and (3) the **LLM backbone**, a Llama 2 7B-parameter large language model [10].

## 3.1 Preliminaries: Vision-Language Models

The architecture of most recent VLMs [20, 42–44] consists of three main parts (see Fig. 2): (1) a **visual encoder** that maps image inputs to "image patch embeddings", (2) a **projector** that visual embeddings and maps them into the input space of a large language model, and (3) a **language model backbone**. During VLM training, the model is trained end-to-end with a next text token prediction objective on paired or interleaved vision and language data curated from various Internet sources.

In this work, we build on the Prismatic-7B VLM [44]. Prismatic follows the same standard architecture described above, with a 600M-parameter **visual encoder**, a small 2-layer MLP **projector**, and a 7B-parameter Llama 2 **language model backbone** [10]. Notably, Prismatic uses a *two-part* visual encoder, consisting of pretrained SigLIP [76] and DinoV2 [33] models. Input image patches are passed separately through both encoders and the resulting feature vectors are concatenated channel-wise. In contrast to the more commonly used vision encoders such as CLIP [77] or SigLIP-only encoders, the addition of DinoV2 features has been shown to be helpful for improved spatial reasoning [44].

## 3.2 OpenVLA Training Procedure

To train OpenVLA, we fine-tune a pretrained Prismatic-7B VLM backbone for robot action prediction (see Fig. 2). We formulate the action prediction problem as a "vision-language" task, where an input observation image and a natural language task instruction are mapped to a string of predicted robot actions [7]. To enable the VLM's language model backbone to predict robot actions, we represent the actions in the output space of the LLM by mapping continuous robot actions to discrete tokens used by the language model's tokenizer. Following Brohan et al. [7], we discretize each dimension of the robot actions separately into one of 256 bins. For each action dimension, we set the bin width to uniformly divide the interval between the $1^{st}$ and $99^{th}$ quantile of the actions in the training data. Using quantiles instead of the min-max bounds Brohan et al. [7] used allows us to ignore outlier actions in the data that could otherwise drastically expand the discretization interval and reduce the effective granularity of our action discretization.

Using this discretization, we obtain N discrete integers $\in [0 \dots 255]$ for an $N$-dimensional robot action. We opt for simplicity and follow Brohan et al. [7]'s approach by simply overwriting the 256 *least used* tokens in the Llama tokenizer's vocabulary (which corresponds to the last 256 tokens) with our action tokens. OpenVLA is trained with a standard next-token prediction objective, evaluating the cross-entropy loss on the predicted action tokens only.

## 3.3 Training Data

The goal in constructing the OpenVLA training dataset is to capture a large diversity of robot embodiments, scenes, and tasks. This enables the final model to control various robots out of the box *and* admits efficient fine-tuning to new robot setups. We leverage the Open X-Embodiment dataset [1] (OpenX) as a base to curate our training dataset. The full OpenX dataset, at the time of writing, consists of more than 70 individual robot datasets, with more than 2M robot trajectories, that were pooled into a coherent and easy-to-use data format in a large community effort. To make training on this data practical, we apply multiple steps of data curation to the raw dataset.

The goals of this curation are to ensure (1) a coherent input and output space across all training datasets, and (2) a balanced mix of embodiments, tasks, and scenes in the final training mixture. To address (1), we follow [1, 5] and restrict our training dataset to contain only manipulation datasets with at least one 3rd person camera and use single-arm end-effector control. For (2), we leverage the data mixture weights of Octo [5] for all datasets that pass the first round of filtering. Octo heuristically down-weights or removes less diverse datasets and up-weights datasets with larger task and scene diversity; see Octo Model Team et al. [5] for details.

We also incorporate multiple datasets into our training mixture that were added to the OpenX dataset since the release of Octo, including the DROID dataset [11], although at a conservative mixture weight of 10%. In practice, we found that the action token accuracy on DROID remained low throughout training, suggesting a larger mixture weight or model may be required to fit it. To not jeopardize the quality of the final model, we removed DROID from the data mix for the final third of training. We provide a complete overview of the used datasets and mixture weights in Appendix A.

### 3.4 OpenVLA Design Decisions

When developing the OpenVLA model, we explored various design decisions in smaller-scale experiments before starting the final model training run. Concretely, we trained and evaluated OpenVLA models on BridgeData V2 [6] for our initial experiments, instead of training on the full OpenX mixture, to increase iteration speed and reduce computational cost. We summarize key learnings from these explorations below.

**VLM Backbone.** Initially, we experimented with multiple VLM backbones. Apart from Prismatic [44], we tested fine-tuning IDEFICS-1 [78] and LLaVA [79] for robot action prediction. We found all three models to be suitable backbones for VLA training, with similar downstream performance in our initial BridgeV2 evaluations. We ultimately chose Prismatic for its improved spatial reasoning capabilities via fused SigLIP-DinoV2 backbones (see Section 3.1).

**Image Resolution.** The resolution of input images has significant impact on the computational requirements of VLA training, since higher-resolution images result in more image patch tokens and thus longer context lengths that quadratically increase training compute. We compared VLAs with $224 \times 224$px and $384 \times 384$px inputs, but found no performance difference in our evaluations, while the latter takes 3x longer to train. We thus opt for a resolution of $224 \times 224$px for the final OpenVLA model. Note that on many VLM benchmarks, increased resolution does improve performance [44, 80, 81], but we did not see this trend (yet) for VLAs.

**Fine-Tuning Vision Encoder.** Prior work on VLMs found that freezing vision encoders during VLM training typically leads to higher performance [44]. Intuitively, a frozen vision encoder may better preserve the robust features learned from its Internet-scale pretraining. However, we found fine-tuning the vision encoder during VLA training to be crucial for good VLA performance. A possible explanation is that the pretrained vision backbone may not capture sufficient fine-grained spatial details about important parts of the scene to enable precise robotic control.

**Training Epochs.** Typical LLM or VLM training runs complete at most one or two epochs. In contrast, we found it important for VLA training to iterate through the training dataset significantly more times, with real robot performance continually increasing until training action token accuracy surpasses 95%. Our final training run completes 27 epochs through its training dataset.

**Learning Rate.** We swept the learning rate across multiple orders of magnitude for VLA training and achieved the best results using a fixed learning rate of 2e-5 (the same learning rate used during VLM pretraining [44]). We did not find learning rate warmup to provide benefits.

## 4 Experiments

The goal of our experimental evaluations is to test OpenVLA's ability to serve as a powerful multi-robot control policy out of the box, as well as be a good initialization for fine-tuning to new robot tasks. Concretely, we aim to answer the following questions: (1) How does OpenVLA compare to prior generalist robot policies, when evaluating on multiple robots and various types of generalization? (2) Can OpenVLA be effectively fine-tuned on a new robot setup and task, and how does it compare to state-of-the-art data-efficient imitation learning approaches? (3) Can we use parameter-efficient

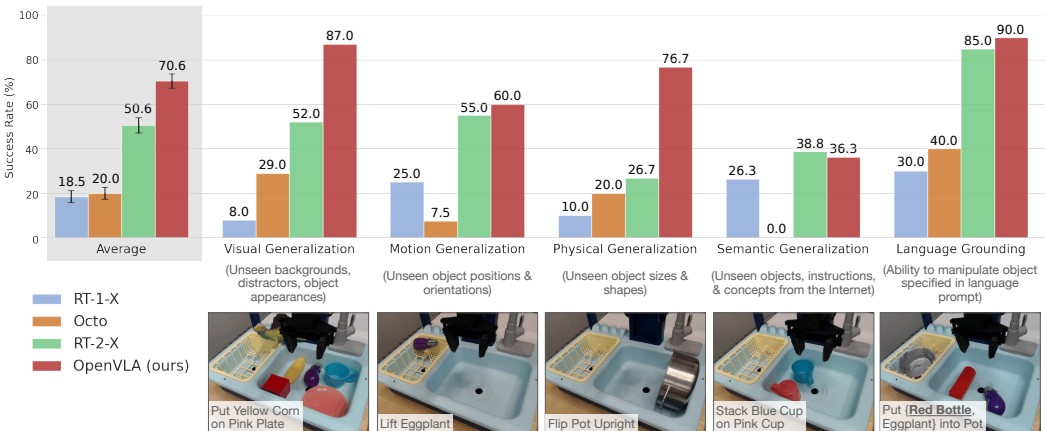

Figure 3: **Bridge V2 WidowX evaluation task categories and results.** We evaluate OpenVLA and prior state-of-the-art generalist robot policies on a comprehensive suite of tasks covering several axes of generalization, as well as tasks that specifically assess language conditioning ability. OpenVLA achieves highest overall performance and even outperforms closed-source model RT-2-X in all categories except for semantic generalization. Average success rates ± StdErr are computed across 170 total rollouts per approach.

fine-tuning and quantization to reduce the computational requirements for training and inference of OpenVLA models and make them more accessible? What are the performance-compute trade-offs?

## 4.1 Direct Evaluations on Multiple Robot Platforms

**Robot Setups and Tasks.** We evaluate OpenVLA's performance "out-of-the-box" on the WidowX robot from the Bridge Data V2 evaluations [6] (see Fig. 1, left) and the mobile manipulator from the RT-1 and RT-2 evaluations [2, 7] ("Google robot"; see Fig. 1, middle). We define a comprehensive set of evaluation tasks in each environment that covers various axes of generalization, such as **visual** (unseen backgrounds, distractor objects, colors/appearances of objects); **motion** (unseen object positions/orientations); **physical** (unseen object sizes/shapes); and **semantic** (unseen target objects, instructions, and concepts from the Internet) generalization. We also assess language conditioning ability in scenes with multiple distractor objects. See Fig. 3 and Fig. 4 for example task images and Appendix B for a detailed description of all tasks. Overall, we evaluated each method with 170 Bridge V2 rollouts and 60 Google robot rollouts, all conducted as A/B evaluations.

**Comparisons.** We compare OpenVLA's performance to three prior generalist manipulation policies: **RT-1-X** [1], **RT-2-X** [1], and **Octo** [5]. **RT-1-X** (35M parameters) and **Octo** (93M parameters) are transformer policies trained from scratch on subsets of the OpenX dataset; Octo is the state of the art among open-source manipulation policies. **RT-2-X** (55B parameters) is a state-of-the-art, closed VLA that leverages Internet-pretrained vision-language backbones.

The results are summarized in Fig. 3 for Bridge V2 evaluations and Fig. 4 for Google robot evaluations (per-task breakdown in Appendix, Table 4 and Table 6). We find that both RT-1-X and Octo struggle on the tested tasks, often failing to manipulate the correct object, especially when distractors are present. RT-2-X clearly outperforms both RT-1-X and Octo, demonstrating the benefits of large, pretrained VLMs for robotics.

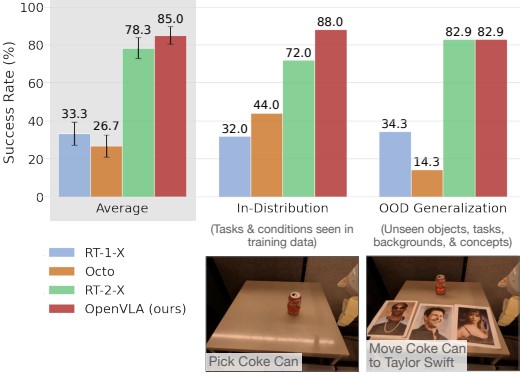

Figure 4: **Google robot evaluation results.** We evaluate generalist robot policies on in-distribution and out-of-distribution (OOD) tasks on the mobile manipulator used in RT-1 and RT-2 evaluations [2, 7]. We find that OpenVLA and RT-2-X attain comparable performance and significantly outperform RT-1-X and Octo overall.

Notably, OpenVLA performs comparably to RT-2-X on Google robot evaluations and significantly outperforms RT-2-X on Bridge V2 evaluations despite being 7x smaller (7B vs. 55B parameters).

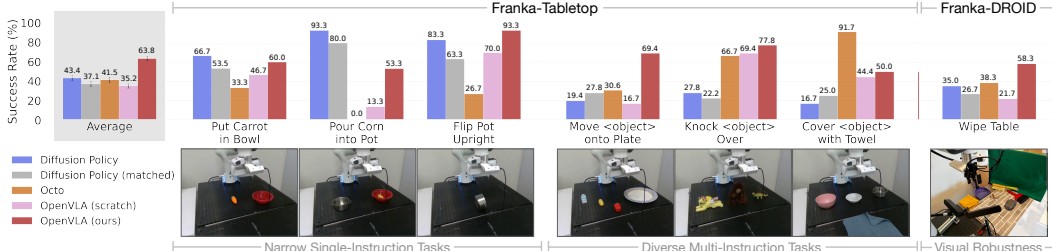

Figure 5: **Fine-tuning to new robot setups.** We fine-tune OpenVLA on 10-100 demonstrations across 7 Franka Emika Panda tasks, ranging from single-instruction tasks to diverse multi-instruction tasks. Diffusion Policy exhibits strong performance on single-instruction tasks, while OpenVLA performs better on diverse fine-tuning tasks with multiple instructions and distractor objects, achieving highest aggregate performance. Mean success ± StdErr computed across 99 and 30 rollouts per approach for Franka-Tabletop and Franka-DROID, respectively.

Qualitatively, both RT-2-X and OpenVLA exhibit markedly more robust behaviors than the other tested models, such as approaching the correct object when distractor objects are present, properly orienting the robot's end-effector to align with the orientation of the target object, and even recovering from mistakes such as insecurely grasping objects.[1] The higher performance of OpenVLA over RT-2-X can be attributed to a combination of factors: we curated a much larger training dataset for OpenVLA with 970k trajectories vs. 350k for RT-2-X; we performed more careful cleaning of the training dataset and, e.g., filter out all-zero actions in the Bridge dataset (see Appendix E for a detailed discussion); and OpenVLA uses a fused vision encoder that combines pretrained semantic *and* spatial features. See Appendix F for ablation analyses of these components.

### 4.2 Data-Efficient Adaptation to New Robot Setups

While prior works mainly focused on evaluating VLAs "out-of-the-box" [1, 7, 16], effective *fine-tuning* of VLAs to new tasks and robot setups is largely unexplored, yet is key for their widespread adoption. In this section, we investigate OpenVLA's ability to be adapted to new robot setups.

**Robot setups and tasks.** We perform full fine-tuning of all OpenVLA parameters, using small datasets with 10–100 demonstrations of a target task (see Fig. 5; we explore parameter-efficient fine-tuning approaches in Section 4.3). We test OpenVLA in two setups: **Franka-Tabletop**, a stationary, table-mounted Franka Emika Panda 7-DoF robot arm; and **Franka-DROID**, the Franka setup from the DROID dataset [11], mounted on a movable standing desk. Franka robot arms are widely used in the robot learning community and thus a likely "target" of OpenVLA fine-tuning.

**Comparisons.** We compare to **Diffusion Policy** [3], a state-of-the-art data-efficient imitation learning approach, trained from scratch. We also compare to **Diffusion Policy (matched)**, a version of Diffusion Policy that matches the input and output specifications of OpenVLA (i.e., no history, no action chunking). Additionally, we evaluate **Octo** [5] fine-tuned on the target dataset (RT-2-X does not support fine-tuning). We also fine-tune OpenVLA on the same target dataset, and the resulting policy is denoted by **OpenVLA**. Finally, as an ablation, we compare to **OpenVLA (scratch)**, which omits OpenX pretraining and directly fine-tunes our base Prismatic VLM on the target robot setup.

We present the results in Fig. 5 (per-task breakdown in Appendix, Table 7). We find that both versions of Diffusion Policy are competitive with or outperform the generalist policies Octo and OpenVLA on narrower single-instruction tasks like "Put Carrot in Bowl" and "Pour Corn into Pot", but the pretrained generalist policies perform better in more diverse fine-tuning tasks that involve multiple objects in the scene and require language conditioning. OpenX pretraining for Octo and OpenVLA enables the models to better adapt to these more diverse tasks where language grounding is important; we see evidence for this in the lower performance of OpenVLA (scratch).

Overall, we find that OpenVLA achieves the highest average performance. It clearly outperforms Octo while being trained on the same robot data, attesting to the benefits of Internet-scale pretraining. Notably, prior works achieve strong performance only in *either* precise *or* diverse tasks, resulting in widely varying success rates. OpenVLA is the only approach that achieves at least 50% success

---

[1]See https://openvla.github.io for videos of qualitative results.

rate across all tested tasks, suggesting that it can be a strong default option for imitation learning tasks, particularly if they involve a diverse set of language instructions. For narrower but highly dexterous tasks, Diffusion Policy still shows smoother and more precise trajectories; incorporating action chunking and temporal smoothing, as implemented in Diffusion Policy, may help OpenVLA attain the same level of dexterity and may be a promising direction for future work.

## 4.3 Efficient OpenVLA Fine-Tuning and Inference

| Strategy | Success Rate | Train Params ($\times 10^6$) | VRAM (batch 16) |
|---|---|---|---|
| Full FT | **69.7 ± 7.2 %** | 7,188.1 | 163.3 GB* |
| Last layer only | 30.3 ± 6.1 % | 465.1 | 51.4 GB |
| Frozen vision | 47.0 ± 6.9 % | 6,760.4 | 156.2 GB* |
| Sandwich | 62.1 ± 7.9 % | 914.2 | 64.0 GB |
| LoRA, rank=32 | **68.2 ± 7.5%** | **97.6** | **59.7 GB** |
| rank=64 | **68.2 ± 7.8%** | 195.2 | 60.5 GB |

Table 1: **Parameter-Efficient Fine-Tuning Evaluation.** LoRA fine-tuning achieves the best performance-compute trade-off, matching full fine-tuning performance while training only 1.4% of the parameters. Mean success ± StdErr across 33 rollouts per approach on select Franka-Tabletop tasks (see Table 8 for details). *: Sharded across 2 GPUs with FSDP [82].

| Precision | Bridge Success | VRAM |
|---|---|---|
| bfloat16 | 71.3 ± 4.8% | 16.8 GB |
| int8 | 58.1 ± 5.1% | 10.2 GB |
| int4 | 71.9 ± 4.7% | 7.0 GB |

Table 2: **Performance with quantized inference.** 4-bit quantization matches the performance of bfloat16 inference (our default approach) while reducing the GPU memory footprint by more than half. Mean success rate ± StdErr computed across 8 representative Bridge V2 tasks [6] and 80 rollouts per approach (see Table 5 for details).

We test various parameter-efficient fine-tuning approaches for OpenVLA[2] across multiple Franka-Tabletop tasks in Table 1: **last layer only** fine-tunes only the last layer of OpenVLA's transformer backbone and the token embedding matrix; **frozen vision** freezes the vision encoder but fine-tunes all other weights; **sandwich fine-tuning** unfreezes the vision encoder, token embedding matrix, and last layer; and **LoRA** uses the popular low-rank adaptation technique of Hu et al. [25] with multiple rank values $r$, applied to all linear layers of the model. We find that fine-tuning of the vision encoder is key (sandwich and LoRA). Importantly, LoRA matches full fine-tuning performance while fine-tuning only 1.4% of the parameters ($r = 32$), enabling us to fine-tune OpenVLA on a new task within 10-15 hours on a *single* A100 GPU – an 8x compute reduction vs. full fine-tuning.

We also test efficient serving of OpenVLA via quantized inference [26, 83]. 8-bit quantization slows down inference across most GPUs due to the overhead of the quantization operations (see Appendix, Fig. 11 for inference speeds on various GPUs). This results in reduced performance when evaluated across 8 representative Bridge V2 tasks (Table 2), as a slower model suffers a distribution shift in system dynamics relative to the training data (see Appendix F.4 for supporting details). 4-bit inference achieves higher throughput due to reduced GPU memory transfer and thus recovers performance of the original bfloat16 model, while requiring less than half the amount of GPU memory.

## 5 Conclusion and Limitations

In this work, we presented OpenVLA, a state-of-the-art, open-source vision-language-action model that obtains strong performance for cross-embodiment robot control out-of-the-box. We also demonstrated that OpenVLA can be easily adapted to new robot setups via parameter-efficient fine-tuning techniques. The current OpenVLA model has several limitations. First, it currently only supports single-image observations; expanding OpenVLA to support multi-image and proprioceptive inputs is an important avenue for future work. Secondly, OpenVLA runs at relatively low control frequencies due to its large size, so improving the inference throughput of OpenVLA is critical to enable VLA control for high-frequency control setups such as ALOHA [84], which runs at 50Hz. Finally, due to compute limitations, many VLA design decisions remain underexplored, such as varying base VLM size, co-training on robot action data and Internet-scale VLM data, and other types of visual features. We hope that the release of the OpenVLA model and codebase will enable the community to jointly investigate these decisions.

---

[2]In Section 4.3, we experiment with a version of the OpenVLA model that is pretrained with a smaller robot data mixture (the same OpenX dataset mixture as Octo) and has a slightly smaller architecture which only uses a SigLIP [76] vision backbone instead of the fused DinoSigLIP encoder. We find that this simpler architecture still achieves strong performance in both fine-tuning tasks and "out-of-the-box" tasks.

**Acknowledgments**

We are grateful to the Toyota Research Institute for providing significant funding and compute resources required to carry out this research. We also thank the Stanford Center for Research on Foundation Models for providing additional compute resources and Google DeepMind for alpha access to the RT-2-X API for our evaluations. We acknowledge additional support from Volkswagen, Physical Intelligence, ONR grants N00014-22-1-2621 and N00014-22-1-2293, the National Science Foundation through IIS-2246811, and DARPA ANSR.

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

# A  Data Mixture Details

We list our used data mixture in Table 3. The mixture mostly follows [5], with a few additional datasets.

| OpenVLA Training Dataset Mixture | |
|---|---|
| Fractal [85] | 12.7% |
| Kuka [45] | 12.7% |
| Bridge[6, 47] | 13.3% |
| Taco Play [86, 87] | 3.0% |
| Jaco Play [88] | 0.4% |
| Berkeley Cable Routing [89] | 0.2% |
| Roboturk [90] | 2.3% |
| Viola [91] | 0.9% |
| Berkeley Autolab UR5 [92] | 1.2% |
| Toto [93] | 2.0% |
| Language Table [94] | 4.4% |
| Stanford Hydra Dataset [95] | 4.4% |
| Austin Buds Dataset [96] | 0.2% |
| NYU Franka Play Dataset [97] | 0.8% |
| Furniture Bench Dataset [98] | 2.4% |
| UCSD Kitchen Dataset [99] | <0.1% |
| Austin Sailor Dataset [100] | 2.2% |
| Austin Sirius Dataset [101] | 1.7% |
| DLR EDAN Shared Control [102] | <0.1% |
| IAMLab CMU Pickup Insert [103] | 0.9% |
| UTAustin Mutex [104] | 2.2% |
| Berkeley Fanuc Manipulation [105] | 0.7% |
| CMU Stretch [106] | 0.2% |
| BC-Z [55] | 7.5% |
| FMB Dataset [?] | 7.1% |
| DobbE [107] | 1.4% |
| DROID [11] | 10.0%[3] |

Table 3: OpenVLA training data mixture using datasets from the Open X-Embodiment dataset [1], following [5] with a few additions.

# B  Evaluation Tasks and Detailed Results

In this section, we provide more details on the BridgeData V2 WidowX and Google robot evaluations discussed in Section 4.1, as well as the Franka-Tabletop and Franka-DROID fine-tuning evaluations discussed in Section 4.2.

## B.1  Bridge V2 WidowX Evaluation Details

Here we focus specifically on BridgeData V2 evaluations discussed in Section 4.1.

### B.1.1  Bridge V2 Evaluation Tasks

As described in Section 4.1, we evaluate each generalist robot manipulation policy on 17 tasks with 10 trials each. In this section, we provide details on the task categories and individual tasks.

In total, we evaluate on 5 visual generalization tasks, 2 motion generalization tasks, 3 physical generalization tasks, 4 semantic generalization tasks, and 3 language grounding tasks. Note that all tasks we evaluate on introduce some form of distribution shift since we are unable to procure the exact objects used in the original dataset (other distribution shifts naturally arise as we reproduce a real-world test environment originally constructed at a different location; see Appendix B.1.2 for a detailed discussion on such distribution shifts). All 17 tasks are depicted in Fig. 6. Each rollout is

---

[3]We remove DROID for the last third of training due to slow learning progress (see Section 3.3) and re-distribute its mixture weights across all other datasets.

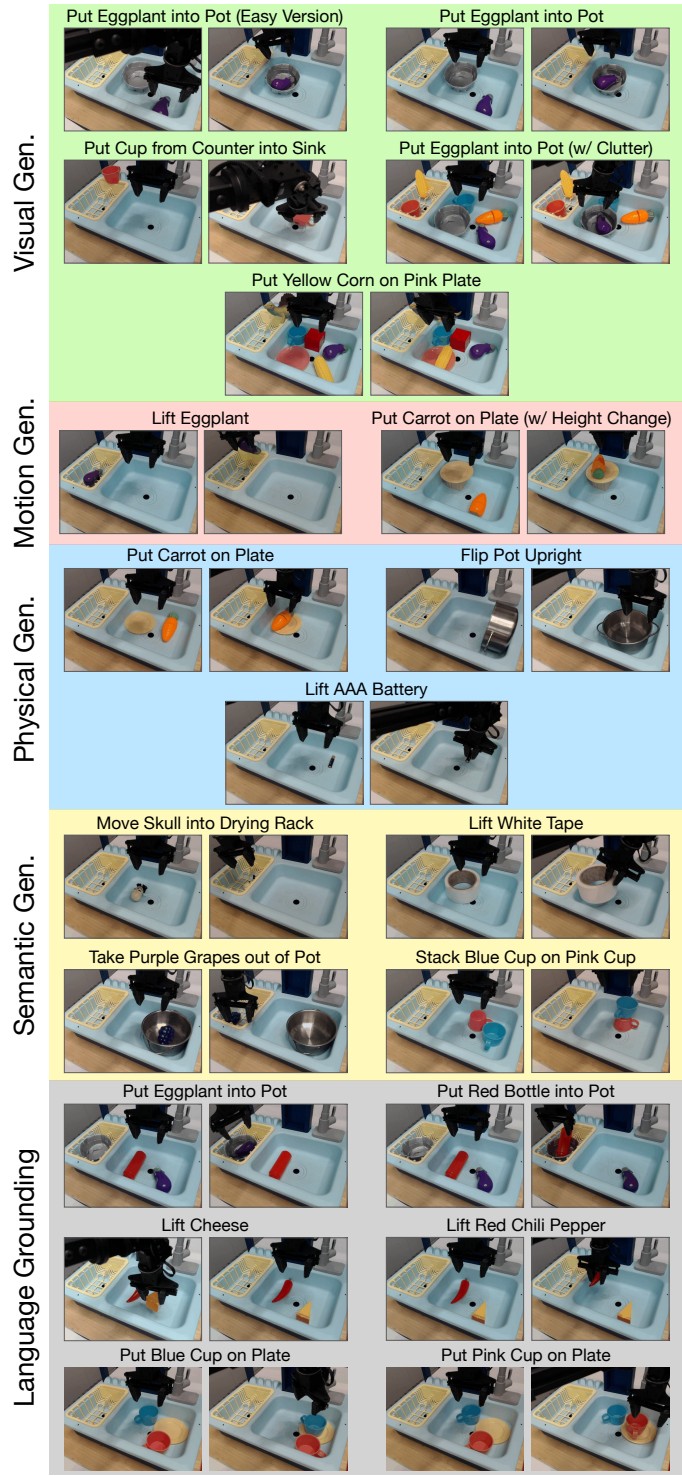

Figure 6: **BridgeData V2 WidowX robot evaluation tasks.** We evaluate every generalist robot policy on 4 types out-of-distribution (OOD) generalization tasks: **visual**, **motion**, **physical**, and **semantic** (as defined in Section 4.1). Every pair of images shows the start state and an example end state after the robot completes the task. We also rigorously assess **language grounding** in the 3 tasks shown in the bottom 3 rows, by changing the prompt while fixing the initial state and testing whether the policy can approach the correct target object.

marked as a failure (0) or success (1). In some more difficult tasks, we record partial successes (0.5); we describe the conditions for partial credit in the task descriptions below.

Below we describe each of the 17 tasks, in the order shown in :

1. **Put Eggplant into Pot (Easy Version)**: The robot's goal is to pick up the eggplant and drop it into the pot. This is a **visual generalization** task because we use a handcrafted paper pot that has a different appearance than the pot used in the original BridgeData V2 training dataset (since we are unable to procure the original pot). Unlike all 16 other tasks, for this particular task we initialize the robot's end-effector directly above the eggplant before rolling out the policy; hence, we call this the "Easy Version" of the "Put Eggplant into Pot" task.

2. **Put Eggplant into Pot**: This is the same task as described above, except that the robot's end-effector is not initialized directly above the eggplant. Instead, we initialize it in a position that is fixed across all rollouts, which means that the robot must horizontally reach for the eggplant first before manipulating it. (Note: The same applies to all other tasks described below.) This is a **visual generalization** task for the same reason as above.

3. **Put Cup from Counter into Sink**: The robot's goal is to pick up the pink cup from either the kitchen countertop or drying rack and place it into the sink on the right. This is a **visual generalization** task because we use a pink cup rather than a blue cup (a blue cup is used in the original BridgeData V2 dataset, but we find that none of the methods we evaluate is able to manipulate it reliably – most likely because the color of the cup blends in with the color of the sink).

4. **Put Eggplant into Pot (w/ Clutter)**: This is the same task as the "Put Eggplant into Pot" task, except that it is more difficult due to the presence of several distractor objects. It is a **visual generalization** task for the same reason discussed in the normal "Put Eggplant into Pot" task, and even more so given unseen distractors in the scene. *Partial credit (0.5 out of 1) is rewarded when the robot moves towards the correct target object.*

5. **Put Yellow Corn on Pink Plate**: The robot's goal is to pick up the yellow corn and place it on the pink plate. This is a **visual generalization** task due to the presence of unseen distractor objects in the scene, such as a green dinosaur on the countertop in the back section of the sink. *Partial credit (0.5 out of 1) is rewarded when the robot moves towards the correct target object.*

6. **Lift Eggplant**: The robot's goal is to grasp and lift the eggplant into the air. This is a **motion generalization** task because the eggplant is initialized in unseen positions and/or orientations, and the robot is forced to move beyond its training distribution of positions and/or orientations and often perform long-range reaching in order to complete the task. (Note: Long-range reaching is not demonstrated in this environment in the original BridgeData V2 demonstrations; see Appendix B.1.2 for details.) We find that this task, though seemingly simple, is deceptively challenging for many policies. *Partial credit (0.5 out of 1) is rewarded when the robot makes contact with the eggplant.*

7. **Put Carrot on Plate (w/ Height Change)**: The robot's goal is to pick up the carrot and place it on the yellow plate. This is a **motion generalization** task because the plate is elevated from its usual position at the bottom of the sink, and the robot must adjust its trajectory to correctly place the carrot on the elevated platform (without knocking down the plate in the process). *Partial credit (0.5 out of 1) is rewarded when the robot grasps the carrot and touches the plate with it.*

8. **Put Carrot on Plate**: This is the same task as above, except that the plate is at its normal position (at the bottom of the sink or drying rack). We consider this a **physical generalization** task because the carrot has a different size and shape than the one used in the original BridgeData V2 dataset, which is shorter and narrower. (Note that the previous version of this task listed above would also technically be a physical generalization task since it involves the same carrot, but we list it under the "motion generalization" category since that is the focus there.)

9. **Flip Pot Upright**: The robot's goal is to manipulate the pot such that it is oriented upright in the sink at the end of the episode. This is a **physical generalization** task because this pot has a different size and shape than the one used in the original BridgeData V2 training demonstrations (the pot we use is wider and shorter).

10. **Lift AAA Battery**: The robot's goal is simply to grasp the AAA battery and lift it up into the air. This is considered a **physical generalization** task because the battery is much smaller and thinner than target objects seen in the BridgeData V2 training demonstrations in this environment; see Appendix B.1.2 for details. (Note that this target object does not exist in the original BridgeData V2 demonstrations in this environment, so this is also an instance of "semantic generalization", but we classify it solely as "physical generalization" since that is the main focus here).

11. **Move Skull into Drying Rack**: The robot's goal is to grasp the skull windup toy and drop it into the yellow drying rack in the left part of the sink. This is a **semantic generalization** task since the skull is an unseen target object (does not appear in the BridgeData V2 training demonstrations).

12. **Lift White Tape**: The robot's goal is to grasp and lift the white roll of tape into the air. This is a **semantic generalization** task since the white tape roll is an unseen target object (does not appear in the BridgeData V2 training demonstrations). (Note that this task may also be considered as "physical generalization" because of its shape being different than the objects seen in the training demonstrations in this environment; most policies struggle to grasp objects with this ring structure, and they often move the robot's end-effector directly into the center region.)

13. **Take Purple Grapes out of Pot**: The robot's goal is to grasp the purple grapes lying inside the steel pot and remove it from the pot (by lifting it out and/or dropping it anywhere outside the pot). This is a **semantic generalization** task because it is an unseen language instruction; the robot has never seen this task in the original BridgeData V2 training dataset.

14. **Stack Blue Cup on Pink Cup**: The robot's goal is to grasp the blue cup and place it securely on top of the pink cup. This is a **semantic generalization** task because it is an unseen language instruction; the robot has never seen this task in this environment in the original BridgeData V2 training dataset. *Partial credit (0.5 out of 1) is rewarded when the robot grasps the blue cup and touches the pink cup with the blue cup.*

15. **Put {Eggplant, Red Bottle} into Pot**: This is a **language grounding** task. The robot's goal is to put the specified target object into the pot. Both the eggplant and red bottle are present in the scene. We conduct paired evaluations: for the same initial state, we prompt the policy to target the eggplant in one episode, and then the red bottle in the next episode. We test each method 5 times with the eggplant and 5 times with the red bottle, using the same set of 5 initial states for both target objects. *Partial credit (0.5 out of 1) is rewarded when the robot moves towards the correct target object.*

16. **Lift {Cheese, Red Chili Pepper}**: This is a **language grounding** task. The robot's goal is to grasp and lift the specified target object. We conduct paired evaluations as described in the task above. *Partial credit (0.5 out of 1) is rewarded when the robot moves towards the correct target object.*

17. **Put {Blue Cup, Pink Cup} on Plate**: This is a **language grounding** task. The robot's goal is to grasp the specified target object and place it onto the plate. We conduct paired evaluations as described in other language grounding tasks. *Partial credit (0.5 out of 1) is rewarded when the robot moves towards the correct target object.*

### B.1.2 Comparing Evaluation Tasks to Original Bridge V2 Training Data

We conduct our evaluations in a sink environment used in the original BridgeData V2 dataset [6]. We reproduce the environment to match the original environment in the BridgeData V2 dataset with rough approximations for the robot's location relative to the sink, as well as the camera's placement

relative to the scene. Given the lack of precise measurements of these positions in the original dataset, we are unable to reproduce the *exact* environment setup, and natural distribution shifts arise due to slightly different robot, sink, and camera placements. In addition, since we evaluate robot policies in a different location than where the training demonstrations were collected from, other natural distribution shifts arise. For example, the lighting conditions and background (e.g., visible areas behind the sink) are inevitably different than what was seen in the training dataset. Furthermore, we are unable to procure the exact set of objects used in the original BridgeData V2 dataset, so there are distribution shifts between the objects used at train time and those used at test time.

Despite all these challenges, we find that certain generalist policies, such as OpenVLA and RT-2-X, can still generalize and perform various tasks fairly reliably "out-of-the-box". Other generalist policies, such as RT-1-X and Octo, can also complete some tasks, though they struggle when tested with more difficult generalization tasks in our BridgeData V2 evaluation suite.

The original BridgeData V2 dataset includes demonstrations of the following seven tasks in this specific sink environment: "Flip Pot Upright", "Put Carrot on Plate", "Put Cup from Counter (or Drying Rack) into Sink", "Put Eggplant into Pot", "Put Knife on Cutting Board", "Put Spoon in Pot", and "Turn Lever Vertical to Front". See Fig. 7 for samples images of all these tasks from the original dataset. Note that all training demonstrations collected in this environment are initialized such that the robot's end-effector is positioned directly above the target object in the beginning of the episode. (However, this is not the case across all environments in the BridgeData V2 dataset; in some other environments, the robot is initialized farther away from the target object, so it must horizontally reach for the object first before manipulating it.)

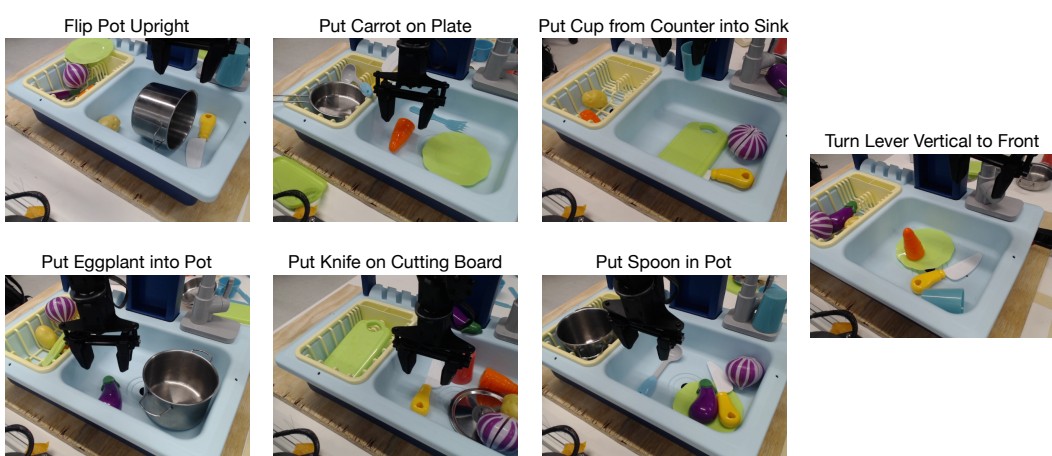

Figure 7: **Original BridgeData V2 sink environment tasks.** Images from sample demonstrations in the sink environment from the original BridgeData V2 dataset reveal that all demonstrations in this environment were initialized such that the robot's end-effector was positioned immediately above the target object. Note that these initial states are different from the initial states we use in our BridgeData V2 evaluation tasks shown in Fig. 6. In our evaluations, we always initialize the robot's end-effector to a fixed location above the sink, rather than positioning it directly above the target object (except for one task: "Put Eggplant into Pot (Easy Version)").

In our BridgeData V2 evaluation suite, only one task – "Put Eggplant into Pot (Easy Version)" – is initialized with the robot's end-effector hovering directly over the target object; in all 16 other tasks, the end-effector is initialized at a fixed location above the sink such that the robot must horizontally reach towards the object. This initial condition, in combination with the distribution shifts we introduce in the various types of OOD generalization in our evaluation suite, challenges the generalist policies and requires a high degree of robustness in order to complete the tasks successfully. Hence, the success rates for policies like RT-1-X and Octo are lower than what is reported in prior works. However, we find that other policies such as RT-2-X and OpenVLA still achieve relatively strong performance despite all these distribution shifts and challenges.

### B.1.3 Detailed Bridge V2 Evaluation Results

See Table 4 for the full BridgeData V2 WidowX evaluation results. The number of successes for each method, out of 10 trials, is listed for each of 17 tasks. OpenVLA achieves strongest performance in the majority of the tasks and has the highest aggregate success rate among the generalist policies. RT-2-X also shows good performance, outperforming RT-1-X and Octo, though it does not perform as well as OpenVLA. RT-1-X and Octo generally experience difficulty in these generalization tasks.

Table 4: **Detailed BridgeData V2 WidowX evaluation results.** We report performance on the full evaluation suite of 17 tasks (discussed in Section 4.1), including visual/motion/physical/semantic generalization tasks and language grounding tasks. Note that *partial success* (score of 0.5) is possible for some tasks; see Appendix B.1.1 for details. We find that OpenVLA performs best in most tasks and achieves highest performance overall, followed by RT-2-X. On the other hand, RT-1-X and Octo struggle in the evaluations, only getting 0–2 successes in several tasks. See Fig. 6 for illustrations of all tasks.

| Category | Task | # Trials | RT-1-X # Successes | Octo # Successes | RT-2-X # Successes | OpenVLA (ours) # Successes |
|---|---|---|---|---|---|---|
| Visual gen | Put Eggplant into Pot (Easy Version) | 10 | 1 | 5 | 7 | **10** |
| Visual gen | Put Eggplant into Pot | 10 | 0 | 1 | 5 | **10** |
| Visual gen | Put Cup from Counter into Sink | 10 | 1 | 1 | 0 | **7** |
| Visual gen | Put Eggplant into Pot (w/ Clutter) | 10 | 1 | 3.5 | 6 | **7.5** |
| Visual gen | Put Yellow Corn on Pink Plate | 10 | 1 | 4 | 8 | **9** |
| Motion gen | Lift Eggplant | 10 | 3 | 0.5 | 6.5 | **7.5** |
| Motion gen | Put Carrot on Plate (w/ Height Change) | 10 | 2 | 1 | **4.5** | **4.5** |
| Physical gen | Put Carrot on Plate | 10 | 1 | 0 | 1 | **8** |
| Physical gen | Flip Pot Upright | 10 | 2 | 6 | 5 | **8** |
| Physical gen | Lift AAA Battery | 10 | 0 | 0 | 2 | **7** |
| Semantic gen | Move Skull into Drying Rack | 10 | 1 | 0 | **5** | **5** |
| Semantic gen | Lift White Tape | 10 | **3** | 0 | 0 | 1 |
| Semantic gen | Take Purple Grapes out of Pot | 10 | **6** | 0 | 5 | 4 |
| Semantic gen | Stack Blue Cup on Pink Cup | 10 | 0.5 | 0 | **5.5** | 4.5 |
| Language grounding | Put {Eggplant, Red Bottle} into Pot | 10 | 2.5 | 4 | **8.5** | 7.5 |
| Language grounding | Lift {Cheese, Red Chili Pepper} | 10 | 1.5 | 2.5 | 8.5 | **10** |
| Language grounding | Put {Blue Cup, Pink Cup} on Plate | 10 | 5 | 5.5 | 8.5 | **9.5** |
| | | Mean Success Rate | 18.5±2.7% | 20.0±2.6% | 50.6±3.5% | **70.6±3.2%** |

Additionally, in Table 5, we provide the full evaluation results for the quantized inference experiments that were summarized in Table 2. For these evaluations, we test policies on 8 representative BridgeData V2 tasks spanning all task categories in the full evaluation suite.

Table 5: **Full quantized inference results.** Here we present the detailed version of the results shown in Table 2.

| Category | Task | # Trials | bfloat16 # Successes | int8 # Successes | int4 # Successes |
|---|---|---|---|---|---|
| Visual gen | Put Eggplant into Pot (Easy Version) | 10 | **9** | 7 | **9** |
| Visual gen | Put Eggplant into Pot | 10 | **7** | **7** | **7** |
| Visual gen | Put Cup from Counter into Sink | 10 | 5 | 3 | **7** |
| Motion gen | Lift Eggplant | 10 | 6 | 4 | **7.5** |
| Physical gen | Put Carrot on Plate | 10 | 6 | 5 | **7** |
| Physical gen | Lift AAA Battery | 10 | **7** | 5 | 3 |
| Semantic gen | Take Purple Grapes out of Pot | 10 | 8 | 8 | **9** |
| Language grounding | Put {Eggplant, Red Bottle} into Pot | 10 | **9** | 7.5 | 8 |
| | | Mean Success Rate | **71.3 ± 4.8%** | 58.1 ± 5.1% | **71.9 ± 4.7%** |

## B.2 Google Robot Evaluation Details

In this section, we provide more details on the Google robot evaluations introduced in Section 4.1.

### B.2.1 Google Robot Evaluation Tasks

On the Google robot, we evaluate each generalist robot policy on 12 tasks with 5 rollouts each, for a total of 60 rollouts. The first five tasks test on in-distribution conditions, and the last seven tasks test on more difficult out-of-distribution (OOD) conditions. All tasks are depicted in Fig. 8. Each rollout is marked as a failure (0) or success (1).

We describe the 12 tasks below:

1. **Pick Coke Can** (in-distribution): The robot is positioned in front of a platform with a can of Coke on top of it. The robot's goal is to grasp and lift the Coke can.

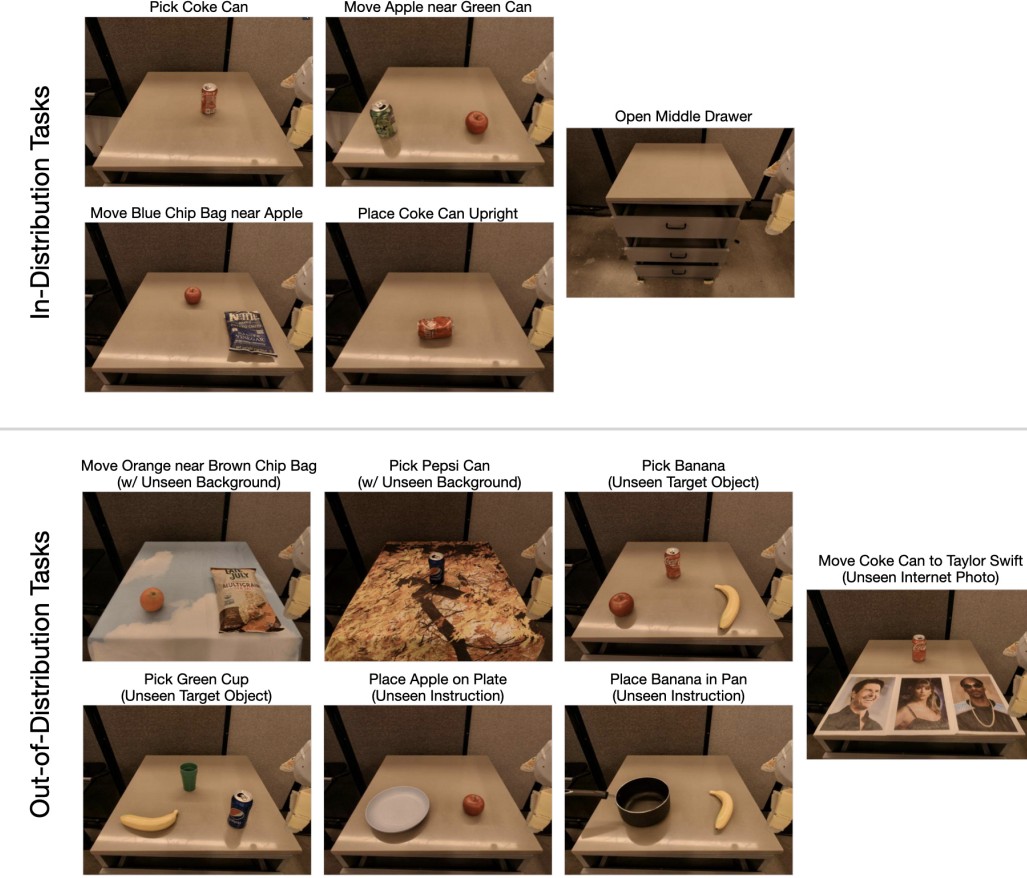

Figure 8: **Google robot evaluation tasks.** We evaluate every generalist robot policy on in-distribution tasks and out-of-distribution (OOD) generalization tasks. OOD tasks involve unseen backgrounds, target objects, instructions/object relations, and semantic concepts (e.g., photos from the Internet that do not appear in robot action data).

2. **Move Apple near Green Can** (in-distribution): The robot is positioned in front of a platform with an apple and a green soda can on top of it. The robot's goal is to grasp the apple and move it next to the green can.

3. **Move Blue Chip Bag near Apple** (in-distribution): The robot is positioned in front of a platform with a blue bag of chips and an apple on top of it. The robot's goal is to grasp the blue bag of chips and move it close to the apple.

4. **Place Coke Can Upright** (in-distribution): The robot is positioned in front of a platform with a can of Coke on top of it, and the can is oriented horizontally on its side. The robot's goal is to grasp the Coke can and orient it to be in a vertical position.

5. **Open Middle Drawer** (in-distribution): The robot is positioned in front of a set of three drawers. The robot's goal is to grasp the middle drawer handle and pull the drawer open.

6. **Move Orange near Brown Chip Bag** (OOD): The robot is positioned in front of a platform with a brown bag of chips and an orange on top of it. A tablecloth with blue sky and white cloud patterns covers the platform underneath the objects. The robot's goal is to grasp the orange and bring it next to the bag of chips. This task is OOD because the orange is an unseen object relative to the training dataset, and the tablecloth is an unseen background.[4]

---

[4]See Appendix of Brohan et al. [7] for a detailed list of OOD conditions in Google robot evaluations.

7. **Pick Pepsi Can** (OOD): The robot is positioned in front of a platform with a can of Pepsi on top of it. A tablecloth with bright yellow/brown patterns covers the platform underneath the can. The robot's goal is to grasp and lift the can. This task is OOD because the Pepsi can is an unseen object, and the tablecloth is an unseen background.

8. **Pick Banana** (OOD): The robot is positioned in front of a platform with an apple, a can of Coke, and a banana. The robot's goal is to grasp and lift the banana. This task is OOD because the banana is an unseen target object.

9. **Pick Green Cup** (OOD): The robot is positioned in front of a platform with a banana, a can of Pepsi, and a green cup. The robot's goal is to grasp and lift the green cup. This task is OOD because all objects in the scene are unseen in the training data.

10. **Place Apple on Plate** (OOD): The robot is positioned in front of a platform with a plate and an apple. The robot's goal is to grasp the apple and move it onto the plate. This task is OOD because it is a novel instruction describing an unseen object relation: training demonstrations only cover moving the apple *near* the plate, rather than placing it *on top of* the plate.

11. **Place Banana in Pan** (OOD): The robot is positioned in front of a platform with a pan and a banana. The robot's goal is to grasp the banana and move it into the pan. This task is OOD because the banana is an unseen target object, and it is a novel instruction describing an unseen object relation, as explained in the previous task.

12. **Move Coke Can to Taylor Swift** (OOD): The robot is positioned in front of a platform with a can of Coke and photos of three different celebrities, including Taylor Swift. The robot's goal is to grasp the can and move it to the photo of Taylor Swift. This task is OOD because the photos of the celebrities are unseen in the robot interaction data.

### B.2.2 Detailed Google Robot Evaluation Results

Table 6: **Detailed Google robot evaluation results.** We report full evaluation results for Google robot evaluations discussed in Section 4.1. Each generalist policy is evaluated with 60 rollouts across 12 tasks, covering both in-distribution and out-of-distribution (OOD) testing conditions. In the bottom row, we report mean success rate $\pm$ StdErr for each policy. OpenVLA and RT-2-X both significantly outperform RT-1-X and Octo overall (we bold the mean success rate for both due to overlapping error bars). See Fig. 8 for illustrations of all tasks.

| Category | Task | # Trials | RT-1-X # Successes | Octo # Successes | RT-2-X # Successes | OpenVLA (ours) # Successes |
|---|---|---|---|---|---|---|
| In-distribution | Pick Coke Can | 5 | **5** | 1 | **5** | **5** |
| In-distribution | Move Apple near Green Can | 5 | 3 | 3 | 3 | **5** |
| In-distribution | Move Blue Chip Bag near Apple | 5 | 0 | 3 | 4 | **5** |
| In-distribution | Place Coke Can Upright | 5 | 0 | 0 | **4** | **4** |
| In-distribution | Open Middle Drawer | 5 | 0 | **4** | 2 | 3 |
| OOD | Move Orange near Brown Chip Bag | 5 | 1 | 2 | **5** | **5** |
| OOD | Pick Pepsi Can | 5 | 3 | 0 | **5** | 4 |
| OOD | Pick Banana | 5 | **5** | 3 | **5** | **5** |
| OOD | Pick Green Cup | 5 | 1 | 0 | **5** | **5** |
| OOD | Place Apple on Plate | 5 | 0 | 0 | **4** | **4** |
| OOD | Place Banana in Pan | 5 | 0 | 0 | 2 | **4** |
| OOD | Move Coke Can near Taylor Swift | 5 | 2 | 0 | **3** | 2 |
| | | Mean Success Rate | 33.3±6.1% | 26.7±5.8% | **78.3±5.4%** | **85.0±4.6%** |

Full results for the Google robot evaluations are shown in Table 6. Overall, we find that RT-1-X and Octo experience difficulty on the evaluation tasks; they are often unable to achieve a single success out of five trials in several tasks. On the other hand, RT-2-X and OpenVLA demonstrate strong performance, completing every task at least two times out of five trials; these two VLA policies perform comparably with each other on this particular evaluation suite.

### B.3 Data-Efficient Adaptation Experiment Details

In this section, we provide more details on the data-efficient adaptation experiments discussed in Section 4.2, where we investigate the effectiveness of fine-tuned OpenVLA policies on new robot setups such as Franka-Tabletop and Franka-DROID.

### B.3.1 Franka-Tabletop and Franka-DROID Tasks

We collect 10–150 demonstrations of each of seven tasks. The first six tasks correspond to a robot setup which we denote as "Franka-Tabletop" (Franka Emika Panda robot mounted on top of a table), and the final task corresponds to a robot setup which we call "Franka-DROID".

In the Franka-Tabletop setup, the first three of six tasks correspond to single-instruction tasks and are narrow, while the last three tasks correspond to multi-instruction tasks in which multiple objects are present in the scene and the robot must manipulate the correct one depending on the language instruction.

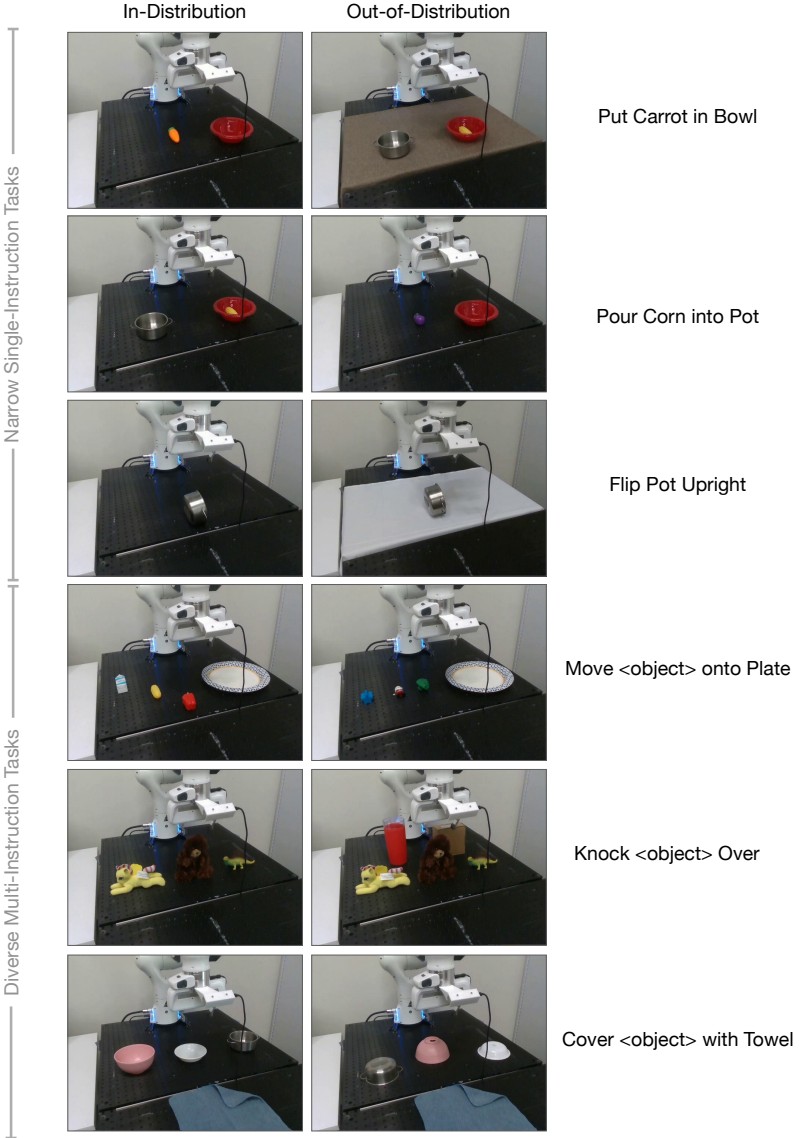

Figure 9: **Franka-Tabletop fine-tuning tasks.** Franka-Tabletop tasks used in the data-efficient adaptation experiments in Section 4.2 and described in detail in Fig. 9 are depicted above. The first three of six tasks, shown in the top three rows, only involve a single instruction, while the last three tasks in the bottom three rows involve multiple objects and instructions (the instructions specify the target object or target location). The first column shows sample initial states matching the training data distribution, while the second column shows out-of-distribution (OOD) initial states (e.g., unseen backgrounds, target objects, distractors, and object positions/orientations). Every policy in Section 4.2 is evaluated with 10–12 rollouts on in-distribution tasks and 5–6 rollouts on OOD tasks.

Below we describe each of the six Franka-Tabletop tasks shown in Fig. 9:

1. **Put Carrot in Bowl** (single-instruction): The robot's goal is to grasp the carrot and place it into the bowl. We collect 50 demonstrations of this task for the training dataset, randomly placing the carrot and the bowl at different locations on the table in every episode. The carrot is always initialized on the left side of the bowl. During evaluation, each trial is recorded as a success (1) or failure (0); there is no partial credit.

2. **Pour Corn into Pot** (single-instruction): The robot's goal is to grasp the red bowl, move towards the steel pot, and pour the contents (a yellow corn) into the pot. We collect 50 demonstrations of this task for the training dataset, randomly placing the bowl and the pot at different locations on the table in every episode. The bowl is always initialized on the right side of the pot. During evaluation, each trial is recorded as a success (1) or failure (0); there is no partial credit.

3. **Flip Pot Upright** (single-instruction): The robot's goal is to grasp the steel pot (which is initially oriented vertically), rotate it to be in the upright position, and place it back onto the table. We collect only 10 demonstrations of this task for the training dataset, randomly placing the steel pot at various locations within a small section of the table. During evaluation, each trial is recorded as a success (1), failure (0), or partial success (0.5). Partial successes include grasping the pot but not orienting it upright, or knocking it over to the upright position but not carefully guiding it. The robot must release the pot at the end of the episode for full credit.

4. **Move <object> onto Plate** (multi-instruction): The robot's goal is to grasp one out of three objects (depending on the target specified in the language instruction) and place it on the plate on the right side of the table. We collect 150 demonstrations of this task for the training dataset, randomly placing different combinations of three objects on the table and selecting one as the target. The plate is always initialized on the right side of the table. During evaluation, each trial is recorded as a success (1), failure (0), or partial success (0.5). Partial success is recorded when the first object that the robot makes contact with is the correct target object (i.e., the object specified in the language instruction), but the robot does not complete the task.

5. **Knock <object> Over** (multi-instruction): The robot's goal is to approach one out of three objects (depending on the target specified in the language instruction) and push it until it falls over. We collect 70 demonstrations of this task for the training dataset, randomly placing different combinations of three objects on the table and selecting one as the target. During evaluation, each trial is recorded as a success (1), failure (0), or partial success (0.5). Partial success is recorded when the first object that the robot makes contact with is the correct target object (i.e., the object specified in the language instruction), but the robot does not complete the task.

6. **Cover <object> with Towel** (multi-instruction): The robot's goal is to grasp the blue towel and place it on one out of three objects (depending on the target specified in the language instruction). We collect 45 demonstrations of this task for the training dataset, randomly placing different combinations of three objects on the table. During evaluation, each trial is recorded as a success (1), failure (0), or partial success (0.5). Partial success is recorded when the first object that the robot touches with the towel is the correct target object (i.e., the object specified in the language instruction), but the robot does not complete the task (e.g., it drops the towel onto the table instead of on top of the target object). Full credit is given when any part of the towel is resting over the top surface of the target object, i.e., the object does not need to be fully covered.

For every Franka-Tabletop task, we evaluate each method with 10–12 in-distribution trials and 5–6 OOD generalization trials. The in-distribution and OOD test conditions are depicted in Fig. 9 (second column).

We describe the OOD test conditions for each of the six tasks below:

1. Put Carrot in Bowl (OOD): An eggplant (unseen object) replaces the carrot.

2. Pour Corn into Pot (OOD): An unseen brown tablecloth covers the tabletop.

3. Flip Pot Upright (OOD): An unseen white tablecloth covers the tabletop

4. Move <object> onto Plate (OOD): A set of three unseen objects are placed on the table.

5. Knock <object> Over (OOD): Two unseen distractor objects (red plastic cup and brown box) are positioned behind the set of three seen objects.

6. Cover <object> with Towel (OOD): The three objects on the table are placed upside-down and at unseen positions.

Finally, in the Franka-DROID environment, we experiment with one task and variants of it: **Wipe Table** (see Fig. 10). In this task, the robot's goal is to grab the brush and sweep all three small brown objects into the dustpan. We collect 70 demonstrations for this task for the training dataset, varying the positions of all the objects.

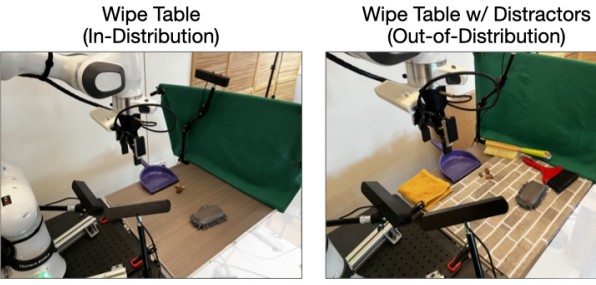

Figure 10: **Franka-DROID fine-tuning task.** The "Wipe Table" task shown here is the final task used in the data-efficient adaptation experiments in Section 4.2. The left image shows the initial conditions for an in-distribution trial. The right image shows an out-of-distribution trial in which unseen distractor objects are present on the table. To fully complete the task, the robot must grab the brush and sweep all three objects into the dustpan.

At test time, we evaluate on in-distribution conditions matching the training data (Fig. 10, left), as well as out-of-distribution (OOD) conditions in which distractor objects are also present in the scene on the table (Fig. 10, right). Since there are various possible outcomes for each trial, we define a scoring rubric as follows: The maximum score for each trial is 2 points. The policy receives the full 2 points if the robot sweeps all three objects into the dustpan. It receives 1 point for successfully sweeping one or two objects into the dustpan. Otherwise, it receives 0 points. We evaluate each policy with 18 in-distribution trials and 12 OOD trials, so each policy receives an aggregate score out of 60 points.

### B.3.2 Detailed Franka-Tabletop and Franka-DROID Evaluation Results

Full evaluation results for both Franka-Tabletop and Franka-DROID evaluations are shown in Table 7. We evaluate the methods discussed in Section 4.2. We find that Diffusion Policy demonstrates strong performance on the single-instruction Franka-Tabletop tasks ("Put Carrot in Bowl", "Pour Corn in Pot", and "Flip Pot Upright"), outperforming other methods. However, OpenVLA (Octo Mix) and Octo achieve higher performance in the more diverse multi-instruction tasks ("Move <object> onto Plate", "Knock <object> Over", and "Cover <object> with Towel"). In the Franka-DROID environment, OpenVLA (Octo Mix) obtains strong results. Overall, we find that OpenVLA (Octo Mix) achieves the highest average performance across both tasks.

Additionally, in Table 8, we show the detailed version of the parameter-efficient fine-tuning experiment results summarized in Table 1. In these experiments, we use a representative subset of two Franka-Tabletop tasks, with both in-distribution and OOD variants: one narrow single-instruction task ("Put Carrot in Bowl") and one diverse multi-instruction task ("Move <object> onto Plate"). We use the same number of training demonstrations used in Section 4.2 (50 and 150, respectively), which is delineated in Appendix B.3.1.

Table 7: **Detailed data-efficient adaptation experiment results.** We report the performance of Diffusion Policy trained from scratch on new robot tasks, as well as generalist policies fine-tuned on the same data. Each policy is tested against both in-distribution and out-of-distribution (OOD) generalization conditions. We find that no single policy performs best on all tasks: Diffusion Policy achieves high success rates on single-instruction tasks, while OpenVLA (Octo Mix) and Octo performs well on diverse multi-instruction tasks. In terms of aggregate performance, however, OpenVLA (Octo Mix) obtains the highest average success rate across both environments.

| | | # trials | Diffusion Policy | Diffusion Policy (matched) | Octo | OpenVLA (scratch) | OpenVLA (Octo Mix) (ours) |
|---|---|---|---|---|---|---|---|
| Franka-Tabletop (5Hz) | "Put Carrot in Bowl" (in-distribution) | 10 | **90.0** | 80.0 | 40.0 | 40.0 | **90.0** |
| | "Put Carrot in Bowl" (OOD) | 5 | 20.0 | 0.0 | 20.0 | 20.0 | **40.0** |
| | "Pour Corn into Pot" (in-distribution) | 10 | **100.0** | 90.0 | 0.0 | 10.0 | 50.0 |
| | "Pour Corn into Pot" (OOD) | 5 | **80.0** | 60.0 | 0.0 | 0.0 | 60.0 |
| | "Flip Pot Upright" (in-distribution) | 10 | **100.0** | 85.0 | 40.0 | 85.0 | 90.0 |
| | "Flip Pot Upright" (OOD) | 5 | 50.0 | 20.0 | 0.0 | **60.0** | 40.0 |
| | "Move <object> onto Plate" (in-distribution) | 12 | 25.0 | 25.0 | 41.7 | 50.0 | **79.2** |
| | "Move <object> onto Plate" (OOD) | 6 | 8.3 | 33.3 | 8.3 | **58.3** | 41.7 |
| | "Knock <object> Over" (in-distribution) | 12 | 33.3 | 25.0 | 83.3 | 62.5 | **87.5** |
| | "Knock <object> Over" (OOD) | 6 | 16.7 | 16.7 | 33.3 | 25.0 | **83.3** |
| | "Cover <object> with Towel" (in-distribution) | 12 | 16.7 | 20.8 | **91.7** | 20.8 | 62.5 |
| | "Cover <object> with Towel" (OOD) | 6 | 16.7 | 33.3 | **91.7** | 0.0 | 41.7 |
| | Average | | 48.5±4.9% | 43.4±4.7% | 43.4±4.4% | 38.9±4.5% | **68.2±4.2%** |
| Franka-DROID (15Hz) | "Wipe Table" (in-distribution) | 18 | 50.0% | 27.8% | 52.8% | 30.6% | **55.6%** |
| | "Wipe Table" + Distractors (OOD) | 12 | 12.5% | 25.0% | 16.7% | 20.8% | **54.2%** |
| | Average | | 35.0±8.0% | 26.7±7.5% | 38.3±8.5% | 26.7±7.5% | **55.0±7.7%** |

Table 8: **Detailed parameter-efficient fine-tuning experiment results.** Here we present the detailed task performance results summarized in Table 1.

| | | # trials | Full FT | Last layer only | Frozen vision | Sandwich | LoRA, r=32 | LoRA, r=64 |
|---|---|---|---|---|---|---|---|---|
| Franka-Tabletop (5Hz) | "Put Carrot in Bowl" (in-distribution) | 10 | 90.0 | 40.0 | 40.0 | 90.0 | 60.0 | 90.0 |
| | "Put Carrot in Bowl" (OOD) | 5 | 40.0 | 0.0 | 40.0 | 0.0 | 60.0 | 40.0 |
| | "Move <object> onto Plate" (in-distribution) | 12 | 79.2 | 33.3 | 50.0 | 75.0 | 75.0 | 62.5 |
| | "Move <object> onto Plate" (OOD) | 6 | 41.7 | 33.3 | 58.3 | 41.7 | 75.0 | 66.7 |
| | Average | | 69.7±7.2% | 30.3±6.1% | 47.0±6.9% | 62.1±7.9% | **68.2±7.5%** | **68.2±7.8%** |

## C  Infrastructure for Training and Inference

The final OpenVLA model is trained on a cluster of 64 A100 GPUs for 14 days, or a total of 21,500 A100-hours, using a batch size of 2048. During inference, OpenVLA requires 15GB of GPU memory when loaded in `bfloat16` precision, i.e., without quantization, and runs at approximately 6Hz on one NVIDIA RTX 4090 GPU (without compilation, speculative decoding, or other inference speed-up tricks). We can further reduce the memory footprint of OpenVLA during inference via quantization, and we demonstrate minimal performance regression with quantized VLA execution in Section 4.3. We report inference speed on various consumer- and server-grade GPUs in Fig. 11. For convenience, we implement a remote VLA inference server to allow realtime remote streaming of action predictions to the robot – removing the need for the robot to have a powerful GPU. We release this remote inference solution as part of our open-source code release (Appendix D).

## D  The OpenVLA Codebase

Along with our model, we will release the OpenVLA codebase, a modular PyTorch codebase for training VLA models (see https://openvla.github.io). It scales from fine-tuning VLAs on individual GPUs to training billion-parameter VLAs on multi-node GPU clusters, and supports modern techniques for large transformer model training such as automatic mixed precision (AMP, PyTorch [109]), FlashAttention [110] and fully sharded data parallelism (FSDP, Zhao et al. [82]). Out of the box, the OpenVLA codebase has full support for training on the Open X-Embodiment dataset, integrates with HuggingFace's [21] `AutoModel` class, supports LoRA fine-tuning [25] and quantized model inference [26, 83]. Everything will be released under a permissive MIT license.

## E  RT-2-X vs. OpenVLA in Bridge V2 Evaluations

In this section, we provide additional details on RT-2-X vs. OpenVLA comparisons in BridgeData V2 evaluations discussed in Section 4.1. As discussed previously, OpenVLA is pretrained on a larger subset of OpenX data than RT-2-X and uses a fused SigLIP-DinoV2 vision backbone rather than a single visual encoder. However, in addition to these factors, we believe that OpenVLA's significant improvement upon RT-2-X specifically in BridgeData V2 evaluations (as shown in Fig. 3) also stems from more careful preprocessing of the Bridge dataset.

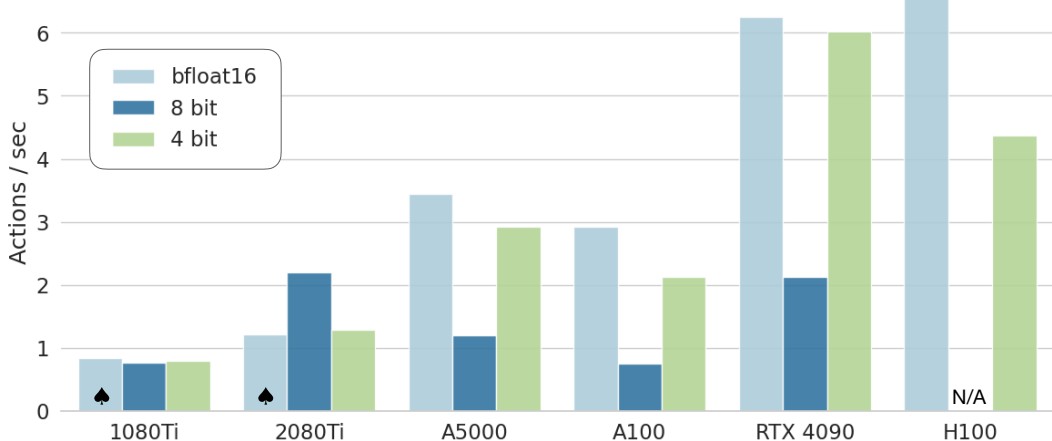

Figure 11: **OpenVLA inference speed for various GPUs.** Both bfloat16 and int4 quantization achieve high throughput, especially on GPUs with Ada Lovelace architecture (RTX 4090, H100). Further speed-ups are possible with modern LLM inference frameworks like TensorRT-LLM [108]. ♠: Model sharded across two GPUs to fit.

During the development of the OpenVLA model, we discovered that the original version of the BridgeData V2 dataset contained many transitions with all-zero (no-op) actions. For instance, in every demonstration, an all-zero action was recorded as the ground-truth action in the first timestep. Consequently, training a highly expressive VLA model on the original dataset without any data preprocessing led to a policy that frequently predicted all-zero actions and froze during evaluations. Therefore, we simply filtered out the first transition in every demonstration when training the OpenVLA model, and this was sufficient for mitigating the freezing behavior in most cases.

However, the RT-2-X model was trained without such data preprocessing, so it often suffers the aforementioned freezing behavior if deployed out of the box without modifying the model querying procedure – which severely deteriorates rollout performance. Since this is a proprietary model that is infeasible for us to re-train (e.g., with our preprocessed version of the BridgeData V2 dataset), we mitigated this issue by simply querying the *second-most-likely* action from the model, since the first-most-likely action was often all zeros while the second-most-likely action was not. (Note that this is the same workaround that was applied by the developers of the RT-2-X model for BridgeData V2 evaluations reported in the Open X-Embodiment experiments [1].) This workaround led to much stronger RT-2-X performance on BridgeData V2 evaluations – though we believe that it is still suboptimal compared to re-training the model on the preprocessed version of the dataset.

We also tried to *dynamically* query RT-2-X, i.e., by first sampling the first-most-likely action and then sampling the second-most-likely action if the first one was all zeros. However, we empirically found that dynamic querying led to worse performance than simply querying the second-most-likely action at all times. We hypothesize that this is due to a change in the robot's dynamics that arises from dynamic querying: pausing in the middle of a trajectory to re-query the model leads to slight interruptions in the robot's movement due to non-neglible latency in the querying pipeline, and this leads to subtle performance degradation. Therefore, we report the performance of RT-2-X when always querying the second-most-likely action, as done in the Open X-Embodiment project [1].

## F   Additional Experiments and Ablations

In this section, we conduct several additional experiments to analyze the effects of individual components of the OpenVLA model architecture and training scheme, as well as provide quantitative evidence for claims made in earlier sections of this work. We aim to answer the following questions:

1. How important is OpenX training and how does it impact OpenVLA's performance (Appendix F.1)?

2. What effect does using a fused SigLIP-DinoV2 vision encoder have on OpenVLA's performance, compared to using a SigLIP-only vision encoder (Appendix F.2)?

3. Is it better to fine-tune or freeze the vision encoder in OpenVLA (Appendix F.3)?

4. How do the quantized inference results discussed in Section 4.3 change when policy performance is disentangled from model inference speed (Appendix F.4)?

We discuss the experimental setup and results addressing each of the above questions sequentially in the following sections.

## F.1 OpenX Training Data Ablation Experiments

As discussed in Section 3.3, OpenVLA is trained on a large dataset of robot embodiments, scenes, and tasks from the Open X-Embodiment dataset [1] (OpenX). In this section, we ablate the OpenX mixture and train a VLA policy solely on one robot dataset, to assess the impact of OpenX training on policy performance. Note that we have already observed the negative effect of ablating OpenX training in the fine-tuning regime, as discussed in Section 4.2 (see OpenVLA (Scratch)), but we discuss additional experiments on another robot embodiment in this section to provide more supporting evidence.

**Experimental setup and tasks.** We compare the original OpenVLA model with **OpenVLA-Bridge**, which is produced by taking the same pretrained VLM as OpenVLA (Prismatic VLM [44]) and fine-tuning it solely on BridgeData V2 [6] rather than the entire OpenX training mixture discussed in Appendix A. We evaluate OpenVLA and OpenVLA-Bridge on a subset of 8 representative tasks from the BridgeData V2 WidowX robot evaluation suite discussed in Appendix B.1.1. The tasks are listed in Table 9.

**Results.** Results for the OpenX training mixture ablation are shown in Table 9. By comparing OpenVLA with OpenVLA-Bridge, we see that performance drops drastically (reduction of 30 percent in absolute success rate), which demonstrates the importance of OpenX pretraining on final policy performance. Although the language grounding performance is not impacted, we observe performance reduction across all generalization categories. This result suggests that the large diversity of scenes, objects, and tasks in the OpenX training mixture is essential for unlocking improved generalization capabilities in the OpenVLA model.

Table 9: **BridgeData V2 WidowX ablation experiment results.** We evaluate various methods on a subset of 8 representative tasks to assess the importance of different components of the OpenVLA model architecture and training scheme. OpenVLA-Bridge is a version of OpenVLA without OpenX training (it is trained only on BridgeData V2), and OpenVLA-Bridge-SigLIP additionally ablates the fused vision backbone by removing the DinoV2 encoder (its vision backbone only consists of the SigLIP encoder). We observe that both OpenX training and the fused vision encoder improve policy performance, though the former has a much greater effect than the latter.

| Category | Task | # Trials | OpenVLA # Successes | OpenVLA-Bridge # Successes | OpenVLA-Bridge-SigLIP # Successes |
|---|---|---|---|---|---|
| Visual gen | Put Eggplant into Pot (Easy Version) | 10 | 10 | 8 | 8 |
| Visual gen | Put Eggplant into Pot | 10 | 10 | 2 | 3 |
| Visual gen | Put Cup from Counter into Sink | 10 | 7 | 4 | 2 |
| Motion gen | Lift Eggplant | 10 | 7.5 | 5.5 | 6.5 |
| Physical gen | Put Carrot on Plate | 10 | 8 | 4 | 1 |
| Physical gen | Lift AAA Battery | 10 | 7 | 2 | 2 |
| Semantic gen | Take Purple Grapes out of Pot | 10 | 4 | 3 | 3 |
| Language grounding | Put {Eggplant, Red Bottle} into Pot | 10 | 7.5 | 8 | 7 |
| | Mean Success Rate | | 76.3 ± 4.8% | 45.6 ± 5.6% | 40.6 ± 5.5% |

## F.2 Dual vs. Single Vision Encoder Experiments

The OpenVLA model architecture consists of a fused vision backbone that combines the SigLIP [9] and DinoV2 [33] encoders. In this section, we ablate the DinoV2 component to assess the importance of using a dual vision encoder.

**Experimental setup and tasks.** We instantiate a model, **OpenVLA-Bridge-SigLIP**, which is a version of OpenVLA that is trained only on BridgeData V2 and consists of only the SigLIP encoder as the vision backbone. We compare this model with the OpenVLA-Bridge model discussed in the previous section (Appendix F.1), which shares the same model architecture as the original OpenVLA

model and is only trained on Bridge robot data. Therefore, the only difference between OpenVLA-Bridge-SigLIP and OpenVLA-Bridge is that the former omits the DinoV2 encoder in the vision backbone. We evaluate these models on the same subset of 8 Bridge tasks described in the previous section.

**Results.** Results for the dual vision encoder ablation are shown in Table 9. The drop in performance from OpenVLA-Bridge to OpenVLA-Bridge-SigLIP implies that additionally including the DinoV2 encoder in the vision backbone improves policy performance. However, the 5 percent reduction in performance here is not as significant as the 30 percent drop in performance observed from ablating OpenX training. The low-level spatial features represented in DinoV2 appear to aid generalization in only some cases.

### F.3   Fine-Tuned vs. Frozen Vision Encoder Experiments

As discussed in Section 3.4, prior work on VLMs observed higher performance from freezing the vision encoder than fine-tuning its parameters [44]. However, when training OpenVLA, we fine-tuned all 7B parameters in the model, including the SigLIP-DinoV2 vision backbone, as we discovered early on during development that fine-tuning the vision encoder led to higher-performing VLAs — a finding which held across various pretrained VLMs and model architectures. We discuss details of such findings below.

**Experimental setup and tasks.** In this section, we report the performance of two VLA policies produced by fine-tuning two different pretrained models from the Prismatic VLMs [44] repository on BridgeData V2. The two pretrained models are named **SigLIP ViT-SO 224px** and **LLaVa v1.5 7B (Reproduction)**; see Karamcheti et al. [44] for details on their architectures and training mixtures. We evaluate both policies on various Bridge tasks shown in Table 10. Note that the evaluation configurations here differ from previously discussed Bridge evaluations, so the results are not directly comparable to results in other similar experiments.

**Results.** Results for the fine-tuned vs. frozen vision encoder experiments are shown in Table 10. We find that for both VLAs tested, fine-tuning the vision encoder leads to significantly higher success rates across various tasks. Qualitatively, in some cases, deploying the frozen vision encoder policies leads to unstable robot behaviors that are clearly suboptimal. Consequently, we decided early on during development to not conduct further experimentation with frozen vision encoders.

Table 10: **Fine-tuned vs. frozen vision encoder experiment results.** We evaluate the performance of fine-tuning ("Fine-Tuned") vs. freezing the vision encoder ("Frozen Vision") in two VLA policies built on top of two different pretrained VLMs from the Prismatic VLMs [44] repository. BridgeData V2 WidowX tasks shown here are performed in the same sink environment used for other Bridge experiments in this work (however, the initial environment configurations here differ, as these evaluations were conducted at an earlier stage in the project). We find that fine-tuning the vision encoder is crucial to obtain good policy performance. Certain frozen vision encoder evaluations were discontinued due to very poor (near-zero) performance and unstable robot behaviors. Among the evaluations where both frozen vision and fine-tuned approaches are tested, fine-tuning the vision encoder leads to 80.0% average success versus 46.7% average success from leaving it frozen.

| | | SigLIP ViT-SO 224px | | LLaVa v1.5 7B (Reproduction) | |
| Task | # Trials | Frozen Vision # Successes | Fine-Tuned # Successes | Frozen Vision # Successes | Fine-Tuned # Successes |
|---|---|---|---|---|---|
| Put Eggplant into Pot | 10 | 7 | 10 | 5 | 9 |
| Put Corn on Plate | 10 | 10 | 9 | 0 | 9 |
| | Mean Success Rate | 85 | **95** | 25 | **90** |
| Put { Eggplant, Red Bottle } into Pot | 4 | 2 | 4 | – | 3 |
| Put { Blue Cup, Pink Cup } on Plate | 4 | 0 | 0 | – | 0 |
| Lift { Cheese, Red Chili Pepper } | 4 | 0 | 3 | – | 2 |
| Put { Strawberry, Lime } into Pot | 4 | 1 | 0 | – | 3 |
| Move { Sushi, Grapes } | 4 | 3 | 4 | – | 3 |
| | Mean Success Rate | 30 | **55** | – | 55 |

### F.4   Additional Quantized Inference Experiments: Disentangling Policy Performance and Model Inference Speed

In Section 4.3, we evaluated OpenVLA with different levels of precision at inference time: half precision (bfloat16), 8-bit quantization, and 4-bit quantization. 8-bit quantization led to lower

BridgeData V2 performance relative to the other two approaches, and we hypothesized that the reduction in performance was caused by lower model inference speed from the operations used in 8-bit quantization. In this section, we conduct experiments to assess the veracity of this claim.

Specifically, we evaluate OpenVLA again with the three different levels of precision listed above, but now with *blocking control*. In other words, each action is fully executed on the robot before the next one is predicted by the policy and executed by the controller. This scheme controls system dynamics across methods with varying amounts of latency and thus allows us to test the quality of a policy's action predictions, independent of its prediction speed. Effectively, the precision levels that have higher throughput – bfloat16 and 4-bit quantization – are forced to run slower to match the dynamics observed when deploying OpenVLA with 8-bit precision. Therefore, we expect OpenVLA's performance with 8-bit precision to match the performance of bfloat16 and 4-bit precision under blocking control.

**Experimental setup and tasks.** We report the performance of OpenVLA with blocking control and quantized inference on the same subset of 8 BridgeData V2 tasks used in Appendix F.1 and Appendix F.2.

**Results.** Quantized inference experiment results with blocking control are shown in Table 11. Unlike in Table 2, where 8-bit quantization led to the worst rollout performance due to low inference speed, here we observe that 8-bit quantization performs comparably to bfloat16 precision and 4-bit quantization given that we evaluate with blocking control to remove the influence of varying inference speeds on task performance. This confirms our hypothesis about the effect of inference speed on 8-bit quantization performance in previous experiments (when using non-blocking control). We also see no substantial performance degradation when using the lowest precision, 4-bit, as also observed in Section 4.3.

Table 11: **Quantized inference experiment results with blocking control.** We report the success rate and standard error of OpenVLA on various BridgeData V2 WidowX tasks with bfloat16 precision (the default approach), 8-bit quantization (int8), and 4-bit quantization (int4) at inference time. All average success rates have overlapping error bars, which suggests that all methods perform comparably.

| Category | Task | # Trials | bfloat16 # Successes | int8 # Successes | int4 # Successes |
|---|---|---|---|---|---|
| Visual gen | Put Eggplant into Pot (Easy Version) | 10 | 10 | 10 | 10 |
| Visual gen | Put Eggplant into Pot | 10 | 9 | 10 | 10 |
| Visual gen | Put Cup from Counter into Sink | 10 | 5 | 5 | 3 |
| Motion gen | Lift Eggplant | 10 | 8 | 7 | 7.5 |
| Physical gen | Put Carrot on Plate | 10 | 10 | 10 | 10 |
| Physical gen | Lift AAA Battery | 10 | 3 | 6 | 4 |
| Semantic gen | Take Purple Grapes out of Pot | 10 | 2 | 2 | 2 |
| Language grounding | Put {Eggplant, Red Bottle} into Pot | 10 | 9 | 9.5 | 8.5 |
| | | Mean Success Rate | 70.0 ± 5.1% | 74.4 ± 4.9% | 68.8 ± 5.2% |

# G   LIBERO Simulation Experiments

Our previous discussions in Section 4.2 and Section 4.3 focused on adapting OpenVLA to novel *real-world* robot setups and tasks. This section explores adapting OpenVLA to *simulated* robot setups and tasks, specifically utilizing the **LIBERO** benchmark [111]. Our experimentation in simulation offers two key advantages:

1. Demonstration of versatility: We show that OpenVLA, despite having been pretrained exclusively on real-world data, can effectively adapt to simulated domains, overcoming potential disparities between real-world and simulated environments and dynamics.

2. Enhanced accessibility and reproducibility: Integration of OpenVLA into a publicly available simulation platform makes our model more accessible to other researchers, especially those who may not have access to robotic hardware. Additionally, simulated experiments are more easily reproduced than their real-world counterparts.

We discuss the experimental setup in Appendix G.1 and the results in Appendix G.2. We release the materials required to reproduce the experiments along with the OpenVLA codebase.

### G.1  LIBERO Simulation Experimental Setup

**Simulation setup and tasks.** The LIBERO benchmark [111] consists of four task suites designed for studying lifelong learning in robotic manipulation, and the original paper therefore investigates both forward and backward transfer to a variety of tasks. In our experiments, we focus solely on supervised fine-tuning on the target task suite, measuring the performance of various policies trained via behavioral cloning on successful demonstrations of the tasks.

We perform experiments with the following four task suites, which each contain 10 tasks with 50 human-teleoperated demonstrations each:

- **LIBERO-Spatial** consists of the same set of objects but different layouts, and tests the model's understanding of spatial relationships.

- **LIBERO-Object** consists of the same scene layouts but different objects, and tests the model's understanding of object types.

- **LIBERO-Goal** consists of the same objects and layouts but different task goals, and tests the model's knowledge of different task-oriented behaviors.

- **LIBERO-Long** (also called **LIBERO-10**) consists of *long-horizon* tasks with diverse objects, layouts, and tasks.

We make the following modifications to each of the training datasets above:

1. To accommodate methods requiring higher-resolution images (such as $256 \times 256$px or $224 \times 224$px), we regenerate all demonstrations at an increased resolution of $256 \times 256$px. Originally, the dataset provided by the benchmark consists of $128 \times 128$px images. We find that simply upscaling these images to $256 \times 256$px results in poor image quality. Therefore, we choose to begin with higher-resolution images, which can be downscaled as necessary, ensuring higher image quality across various resolution requirements. These higher-resolution images were obtained by stepping through the simulation environments with the actions stored in the provided human-collected demonstrations and saving the images rendered by the simulator.

2. We filter out all "no-op" actions from the dataset, i.e., actions that have near-zero magnitude in the translation and rotation components and do not change the state of the robot's gripper. We find that this simple data cleaning step is crucial for highly expressive single-step policies such as OpenVLA, which otherwise learn to imitate these no-op actions and consequently freeze indefinitely at certain states during evaluation.

3. We rotate all third-person images at both train and test time by 180 degrees because we observe that the LIBERO environments return images that are upside down on our hardware.

4. Since we train policies via imitation learning, which expects demonstrations to be successful, we replay all demonstrations in the corresponding simulation environments and filter out the demonstrations that fail to complete the task (as determined by the environments' success criteria). As a result, we remove 68 of 500 LIBERO-Spatial demonstrations, 46 of 500 LIBERO-Object demonstrations, 72 of 500 LIBERO-Goal demonstrations, and 121 of 500 LIBERO-Long demonstratinos.

5. For all methods in our comparisons, we only utilize the static third-person camera images; we do not use the wrist camera images that are additionally provided in the original datasets. This is for sake of having fair comparisons, as OpenVLA's visual inputs only consist of third-person camera images.

**Comparisons.** The methods that we compare include **Diffusion Policy**[5] [3] trained from scratch, **Octo** [5] fine-tuned on the target dataset, and **OpenVLA** fine-tuned on the target dataset via LoRA ($r = 32$) as described in Section 4.3. Each policy is trained independently on each of the task suites

---

[5]We use the implementation of Diffusion Policy that is described in the DROID dataset paper [11], which conditions action generation on DistilBERT [112] language embeddings of the task label.

above (rather than training a single policy on all four suites combined). All policies are trained with the same set of demonstrations, so all methods benefit from the data cleaning steps described above.

**Evaluation details.** To ensure lower variance in the experimental results, all methods are evaluated across 500 trials for each task suite, and the reported performance is the average success rate over three random seeds (resulting in 1500 total trials per statistic). Although we modify the training datasets, as described earlier, we do not change the test environments but rather use the same initial environment configurations provided by the original LIBERO benchmark.

## G.2 LIBERO Simulation Experimental Results

We present the LIBERO experimental results in Table 12. Importantly, we observe that OpenVLA can be effectively adapted to tasks in the LIBERO simulation environments, as it obtains highest average success rate and rank among the tested methods. However, we find that the overall margin between OpenVLA and the other methods are tighter here than in the real-world fine-tuning experiments discussed in Section 4.2. We attribute this to the fact that OpenVLA was pretrained with purely real-world robot data and no simulation data, which suggests that fine-tuning the model on simulated robot tasks may not be as effective as fine-tuning it on real-world tasks due to the domain gap between simulated and real-world environments and dynamics. We see evidence for this notion in the results obtained by Octo – another policy pretrained on large amounts of real-world robot data – which also only achieves a small boost in overall performance relative to a simple, strong baseline such as Diffusion Policy trained from scratch. We expect increased gains in performance for the pretrained and fine-tuned methods if simulation data is added to the pretraining data mixture.

Table 12: **LIBERO simulation benchmark results.** We report the success rate (SR) and standard error of each method for the four task suites in the LIBERO benchmark, averaged over three random seeds with 500 trials each. In addition, we show the ranking of each method within each task suite, where a rank of 1 indicates the strongest method in the suite and a rank of 3 indicates the weakest method. (The average ranking is important to note since it informs which method may be most suitable to use as a default for a variety of tasks; it is more informative than the average success rate, which is not normalized by individual task suite difficulty.) Overall, we find that fine-tuned OpenVLA achieves highest average success rate and rank, followed by fine-tuned Octo and then Diffusion Policy trained from scratch.

| | LIBERO-Spatial | | LIBERO-Object | | LIBERO-Goal | | LIBERO-Long | | Average | |
|---|---|---|---|---|---|---|---|---|---|---|
| | SR (↑) | Rank (↓) | SR (↑) | Rank (↓) | SR (↑) | Rank (↓) | SR (↑) | Rank (↓) | SR (↑) | Rank (↓) |
| Diffusion Policy from scratch | $78.3 \pm 1.1\%$ | 3 | $\mathbf{92.5 \pm 0.7\%}$ | 1 | $68.3 \pm 1.2\%$ | 3 | $50.5 \pm 1.3\%$ | 3 | $72.4 \pm 0.7\%$ | 2.5 |
| Octo fine-tuned | $78.9 \pm 1.0\%$ | 2 | $85.7 \pm 0.9\%$ | 3 | $\mathbf{84.6 \pm 0.9\%}$ | 1 | $51.1 \pm 1.3\%$ | 2 | $75.1 \pm 0.6\%$ | 2 |
| OpenVLA fine-tuned (ours) | $\mathbf{84.7 \pm 0.9\%}$ | 1 | $88.4 \pm 0.8\%$ | 2 | $79.2 \pm 1.0\%$ | 2 | $\mathbf{53.7 \pm 1.3\%}$ | 1 | $\mathbf{76.5 \pm 0.6\%}$ | $\mathbf{1.5}$ |

