# OpenReview forum: "OpenVLA: An Open-Source Vision-Language-Action Model"
_robot-learning.org/CoRL/2024/Conference — CoRL 2024_

### Official Review · Reviewer_LnaQ · 2024-07-19
**Review of Submission 671**

**Originality:** 4
**Technical Quality:** 4
**Clarity Of Presentation:** 4
**Potential Impact:** 4
**Recommendation:** 4
**Confidence:** 3

**Review:**

Strengths:
- The release of the OpenVLA model, along with the training data, code, and fine-tuning notebooks, significantly contributes to the research community, promoting transparency and reproducibility.
- In the experiment, OpenVLA achieves state-of-the-art performance despite having a 7x smaller model than RT-2-X. Furthermore, the model shows generalization capability across various tasks and environments.
- This paper presents parameter-efficient fine-tuning of OpenVLA. It shows that the model can be fine-tuned on only a single A100 GPU utilizing LoRA, and this parameter-efficient fine-tuning is comparable with full fine-tuning.

Weakness and suggestions:
- Although the paper presents extensive evaluations, additional comparisons with more varied robot setups and tasks could provide a more comprehensive assessment of the model's capabilities.
- A discussion regarding the differences in robot body shapes is also important. How well can OpenVLA adapt to different robot shapes using fine-tuning?
- How much data is used in the experiment in section 4.3?

**Quality Of The Limitations Section:**

3

**Questions For Rebuttal:**

See above.

**Robotics Focus:**

4

**Summary Of Paper:**

This paper introduces OpenVLA, a 7B-parameter open-source Vision-Language-Action (VLA) model, trained on 970k robot episodes. The model aims to obtain generalizable policies of multiple robots and tasks for visuomotor control.

**Summary Of Recommendation:**

I highly evaluate this paper.

---

### Official Review · Reviewer_t3E9 · 2024-07-20
**Review for OpenVLA**

**Originality:** 4
**Technical Quality:** 4
**Clarity Of Presentation:** 3
**Potential Impact:** 3
**Recommendation:** 4
**Confidence:** 4

**Review:**

# Strengths

- The paper is clearly written and the authors explored many design choices on lower scale dataset (BridgeData V2)
- OpenVLA model showed better or comparible generalization performance than previous larger closed source model (RT-2-X) on different diverse set of tasks that covers different types of generalization
- In addition, authors showed that the same "generalist" model could be farther finetuned to perform a particular tasks and showed competitave perfomrance to Diffusion policy trained to solve this task. In addition, they also studied the efficient finetuning strategies like LoRA showing that it is possible to finetune their model using small scale resources (e.g. single A100) making the model more accessible for researchers.
- The authors release code and pretrained models thus making usage of VLA more accesible and easier.

# Weaknesses / Questions

- In some cases more detailed results would be beneficial for the community. For example authors stress the importance of vision encoder fine-tuning however are not providing the comparison between freezed and fine-tuned vision encoder. In addition, it would be great to now if the fine-tuning of all the layers is required, or it is enough to fine-tune only last layers of the vision encoder. Similarly it would be great to have comperision with only SigLIP features to see the advantanges of using combination.  It would be great to have those comparisons including absolute numbers of the performance in results in the appendix.
- For fine-tuning to a particular task (Data-Efficient Adaptation to New Robot Setups) is would be great to additionally compare how much more data is needed for Diffusion policy to achieve similar performance (e.g. if with the tasks that require language and differentiating between objects could be solved with more data) this would give an estimate of how much more data efficient is OpenVLA fine-tuning if contrast to training from scratch Diffusion model.

**Quality Of The Limitations Section:**

3

**Questions For Rebuttal:**

# Minor comments/questions:

- In table 2 (Bridge Success) it would be great to split the sources of the decreased performance, for example by testing on some of the task in the simulation and see to what extent the problem in int8 is not appearing there. Also, it would be great to do even more extreme quantization to show the limits of such approach. Besides, it would be useful to include original performance for not quantized model in the same table

> "Neither SigLIP / DinoV2, nor Llama 2 release details about their training data, which likely consists of trillions of tokens of Internet-sourced image-text and text-only data respectively."
- DINOv2 is not using image-text pair but only images so the claim could be more careful about this.

**Robotics Focus:**

4

**Summary Of Paper:**

The authors proposed a novel approach called OpenVLA an open-source vision language action (VLA) model that uses large scale robotics dataset to finetune a pre-trained VML model for robotics tasks.

**Summary Of Recommendation:**

While it would be great to adress current weak points of the paper, overall OpenVLA would be useful in empovering many robotics researcher to start from robotics foundational model and build on top of it.

---

### Official Review · Reviewer_GEDn · 2024-07-20
**The review of OpenVLA**

**Originality:** 4
**Technical Quality:** 4
**Clarity Of Presentation:** 5
**Potential Impact:** 4
**Recommendation:** 4
**Confidence:** 4

**Review:**

**Summary**:

This paper proposes an open-sourced Vision Language Action model which generates discrete action tokens auto regressively. The model is based on a pretrained vision-conditioned language model, and finetuned on a filtered 970K robotic rollouts from the Open-X Embodiment dataset. The vision encoder is consists of a DinoV2 and a SigLP and contains multi-level visual features. Sucessive experiments has demonsrated the effectiveness and efficiency of the model.

**Strength**:
1. The model has a low training consume by utilizing only robotic data and surpass the SOTA end2end models like RT-2-X.

2. The model supports parameter efficient training and inference on consumer gpus, which is a big step forward compared with RT-2.

3. The authors implement serval ablation studies including different VLM backbones and different resolutions.

2. All the code and data are open-sourced, which contributes a lot to the community.

**Weakness**:
1. The authors may test the proposed model over simulator results for fair comparison with other existing works.

2. The results in the ablation study (discussion of OpenVLA) should be released in appendix.

**Quality Of The Limitations Section:**

3

**Questions For Rebuttal:**

See weakness.

**Robotics Focus:**

4

**Summary Of Paper:**

This paper proposes an open-sourced Vision Language Action model which generates discrete action tokens auto regressively. The model utilizes only robotic data and surpass the SOTA end2end models like RT-2-X. All the code and data are released.

**Summary Of Recommendation:**

This paper proposes an open-sourced Vision Language Action model which generates discrete action tokens auto regressively. The model utilizes only robotic data and surpass the SOTA end2end models like RT-2-X. All the code and data are released.

---

### Author Rebuttal · Authors · 2024-08-13

The rebuttal .zip file includes a revised manuscript (PDF) that includes all additional experiments and paper revisions.

---

### Decision · Program_Chairs · 2024-09-04

**Decision:**

Accept

**Comment:**

Strengths:
+ OpenVLA achieves compelling performance over RT-2-X despite being smaller.
+ Code, data, and checkpoints are open-sourced.
+ The paper is written clearly.
+ The experimental setup is thorough.
+ A parameter-efficient fine-tuning process is provided.

Weaknesses:
- No simulation results for fair comparison to existing works.
- Experiments could include: more varied robot setups (different manipulators?), comparison between frozen and fine-tuned vision encoder, ablation with only SigLIP features.

Post rebuttal:
The authors have addressed the concerns. In general, the paper received very positive feedback.